# Description and validation of an intermediate complexity model for ecosystem photosynthesis and evapo-transpiration: ACM-GPP-ETv1

Thomas Luke Smallman[1,2] and Mathew Williams[1,2]

[1]School of GeoSciences, University of Edinburgh, Edinburgh UK
[2]National Centre for Earth Observations, University of Edinburgh, UK

**Correspondence:** T.L. Smallman, t.l.smallman@ed.ac.uk

**Abstract.** Photosynthesis (gross primary production, GPP) and evapo-transpiration (ET) are ecosystem processes with global significance for climate, the global carbon and hydrological cycles, and a range of ecosystem services. The mechanisms governing these processes are complex but well understood. There is strong coupling between these processes, mediated directly by stomatal conductance and indirectly by root zone soil moisture content and its accessibility. This coupling must be effectively modelled for robust predictions of earth system responses to global change. Yet, it is highly demanding to model leaf and cellular processes, like stomatal conductance or electron transport, with responses times of minutes, over decadal and global domains. Computational demand means models resolving this level of complexity cannot be easily evaluated for their parameter sensitivity, nor calibrated using earth observation information through data assimilation approaches requiring large ensembles. To overcome these challenges, here we describe a coupled photosynthesis evapo-transpiration model of intermediate complexity. The model reduces computational load and parameter numbers by operating at canopy scale and daily time steps. Through the inclusion of simplified representation of key process interactions it retains sensitivity to variation in climate, leaf traits, soil states and atmospheric $CO_2$. The new model is calibrated to match the biophysical responses of a complex terrestrial ecosystem model (TEM) of GPP and ET through a Bayesian model-data fusion framework. The calibrated ACM-GPP-ET generates unbiased estimates of TEM GPP and ET, and captures 80-95 % percent of the sensitivity of carbon and water fluxes by the complex TEM. The ACM-GPP-ET model operates three orders faster than the complex TEM. Independent evaluation of ACM-GPP-ET at FLUXNET sites, using a single global parameterisation, shows good agreement with typical $R^2$ ~0.60 for both GPP and ET. This intermediate complexity modelling approach allows full Monte Carlo based quantification of model parameter and structural uncertainties, global scale sensitivity analyses for these processes, and is fast enough for use within terrestrial ecosystem model-data fusion frameworks requiring large ensembles.

# 1   Introduction

Ecosystem photosynthesis and evaporation are key ecosystem fluxes, and their strong coupling generates important feedbacks between plant carbon and water cycles (Tuzet et al., 2003; Bonan and Doney, 2018). Ecosystem photosynthesis, or gross primary productivity (GPP) is generally the sole input of organic carbon into terrestrial ecosystems, ultimately determining potential carbon accumulation rates. Ecosystem evaporation, or evapo-transpiration (ET), is the combination of plant mediated transpiration, soil surface evaporation and subsequent evaporation of rainfall intercepted by plant canopies. The dominant abiotic factors governing the magnitude and variability of GPP are temperature, absorbed photosynthetically active radiation (PAR) and $CO_2$ which are strongly impacted by  leaf area index (LAI). Access to $CO_2$ is controlled via leaf stomata, which provide the primary coupling point between GPP and the water cycle. Stomatal opening results in water loss via transpiration creating a dependency on accessible soil moisture, which is controlled by root biomass and its distribution through the soil profile. Thus, the soil-root interface is a second coupling point between the plant carbon and water cycles (Beer et al., 2009; Bonan and Doney, 2018). State-of-the-art terrestrial ecosystem models (TEMs) provide a mechanistic / process-oriented representation of the coupling between plant carbon and water cycles (e.g. Krinner et al., 2005; Oleson et al., 2010; Smallman et al., 2013; Harper et al., 2016) at leaf or even sub-leaf scale, resolving radiative transfer, stomatal conductance and electron transport. TEMs represent state-of-the-art knowledge on how ecosystems function, and are used to provide meaningful predictions of the responses by and feedbacks from the terrestrial land surface in response to changes in the Earth system (Bonan and Doney, 2018). Mechanistic models linking leaf-level photosynthesis (e.g., Farquhar and von Caemmerer, 1982; Collatz et al., 1991) and transpiration (e.g., Monteith, 1965) through models of stomatal regulation (Medlyn et al., 2011; Williams et al., 1996; Bonan et al., 2014) are well established. Scaling from leaf to canopy scale has grown increasingly complex as the role of non-linear within-canopy variation of both abiotic (e.g., light, temperature, momentum, $CO_2$ and $H_2O$) and biotic (i.e. plant traits) factors on plant carbon-water relations has improved (e.g., Wang and Leuning, 1998; Buckley et al., 2013; Sun et al., 2014; Way et al., 2015; Coble et al., 2016; Scartazza et al., 2016; Nolan et al., 2017; Bonan et al., 2018).

However, the increasing complexity of TEMs presents new challenges. Many of the most complex TEMs are too slow for use in model-data fusion analyses which are reliant on massive ensemble simulations (e.g., Ziehn et al., 2012; Smallman et al., 2017). While effective and more computationally efficient alternative model-data fusion approaches are available they often reply on model code modifications, such as the creation of the model adjoint in variational approaches (e.g., Kuppel et al., 2012; Raoult et al., 2016), or model emulation often resulting in larger uncertainties in their posterior analysis (e.g., Fer et al., 2018).  The complexity of typical TEMs generally prevents a robust quantification of their uncertainties; it is very challenging computationally to determine the sensitivities of TEM model outputs to parameter variation. This hinders interpretation of model-data mismatch. Finally, there are major challenges in procuring sub-daily meteorological observations needed to drive TEMs away from meteorological stations - this is a particularly acute problem in tropical regions. Thus, TEMs are generally run using statistical down-scaled climate reanalysis data, which contain errors The uncertainty generated when these errors are propagated into TEM GPP and ET estimates is comparable to IC model error associated with simulating daily fluxes directly (Williams et al., 1997, 2001a). Thus, IC models have been shown to have similar errors to TEM models but at lower

computational cost (i.e. 1 time versus 24 time steps). Thus, there is considerable value in having less complex, fast-running models that simulate GPP and ET. The challenge here is to produce a model *both* sufficiently mechanistic to represent the coupling between plant carbon and water cycles linking to ecophysiological processes and observations of key global unknowns (e.g. rooting depth), but also computationally fast enough to be integrated into model-data fusion schemes and to allow a full
exploration of parameter-related uncertainties.

Photosynthesis is often estimated using physiologically realistic light, $CO_2$ and temperature response functions (e.g., Jones, 1992; Williams et al., 1997). Evaporation is frequently estimated using simplified versions of the Penman-Monteith model, typically modelling plant stomatal regulation as a function of environmental drivers (e.g., Priestley and Taylor , 1972; Fisher et al., 2008). The impact of moisture limitations on both GPP and ET is commonly achieved through the use of VPD as a proxy
(e.g., Mu et al., 2011; Wang et al., 2017) or using a single soil layer "bucket" (e.g., Martens et al., 2017). While simple models can show skill when compared to in-situ estimates (Mu et al., 2011; Bloom and Williams , 2015; Martens et al., 2017; Wang et al., 2017). They usually estimate a single process, either photosynthesis or evapo-transpiration, neglecting their coupling. Without coupling, the feedbacks between C and water cycles will not be modelled robustly. For instance, there is a high risk that independently calibrated, simple GPP and ET models that are coupled naively in a plant-soil model framework will misdiagnose
the sensitivity of water use efficiency (C fixed per water transpired) and have low predictive capability outside of the calibrated range (e.g., big leaf vs multiple leaf canopy; Tuzet et al., 2003; Wang and Leuning, 1998). Thus, connecting a series of simple models to generate a model of intermediate complexity (IC) carries significant risks. The IC model must represent process interactions effectively. A key test therefore is that any IC model must reproduce the sensitivities of key processes (i.e. GPP, ET), their interactions (WUE) and soil moisture status demonstrated by the state-of-the-art TEMs to ensure flux estimates are
right for the *right* reasons.

This study builds on two previously developed aggregated canopy models (ACM) for GPP (Williams et al., 1997) and ET (Fisher et al., 2008), and an existing state-of-the-art TEM SPA (Williams et al., 1996; Smallman et al., 2013). ACM-GPP simulated daily GPP sensitive to canopy nitrogen (N), temperature, absorbed shortwave radiation and atmospheric $CO_2$ concentration; based on physiologically realistic relationships but lacking a representation of the impact of soil moisture availability on
photosynthesis. Despite this limitation ACM-GPP has been coupled to the DALEC C-cycle model (Williams et al., 2005) and successfully used in model-data fusion experiments to improve our understanding of ecosystem C status, C allocation and residence times (Fox et al., 2009; Bloom and Williams , 2015; Bloom et al., 2016; Smallman et al., 2017) but also carbon-nitrogen interactions (Thomas and Williams et al., 2014). In addition to lacking a soil moisture response on photosynthesis ACM-GPP limits the capacity of DALEC analyses to constrain the root component of the C cycle as roots currently play no ecological role
within the modelling system (i.e. water or nutrient uptake). ACM-ET simulates the bulk ecosystem evapotranspiration based on a modified Penman-Monteith approach sensitive to absorbed shortwave radiation, temperature, vapour pressure deficit and wind speed. However, ACM-ET's bulk approach does not allow for distinguishing between different evaporative sources (i.e. soil surface, root extracted and canopy intercepted rainfall). Thus, it does not account for the different biotic and abiotic drivers which have varied responses to environmental change (Wei et al., 2017). Moreover, ACM-ET does not have a mechanistic cou-
pling to water supply governed by root biomass and root vertical distribution. ACM-GPP and ACM-ET use different empirical

models linking LAI, minimum tolerated leaf water potential and meteorological drivers to estimate canopy conductance. Both ACM's can be calibrated to provide useful GPP and ET estimates, however when combined their predictive capacity for emergent properties such as WUE is limited ($R^2 < 0.2$; data not shown) highlighting the need for further development to reproduce emergent ecosystem properties.

Here we describe a process model of intermediate complexity, ACM-GPP-ET version 1, that simulates gross primary productivity and evapotranspiration. ACM-GPP-ET is a fast, coupled representation of plant carbon and water cycles at ecosystem scale and daily time resolution. Coupling is achieved via a canopy stomatal model that determines $CO_2$ and $H_2O$ exchanges in the canopy. With fewer parameters than many state-of-the-art process orientated models, our model is simpler to calibrate. With a daily time-step and single canopy layer the model is fast, and therefore viable for ensemble modelling. In fact ACM-GPP-ET

is explicitly intended as a replacement for ACM-GPP as part of the DALEC model addressing current weaknesses in simulating carbon-water interactions with our model-data fusion framework. To ensure its realism, ACM-GPP-ET is an emulation of a more complex LSM, SPA (Williams et al., 1996) that resolves leaf scale, hourly exchanges of $CO_2$ and water. SPA also includes a detailed, multi-layer representation of radiative transfer, energy balance, carboxylation and plant-soil interactions at sub-daily timescales. SPA explicitly couples available supply of water from the soil (determined as a function of soil charac-

teristics, root biomass and structure) to demand by the atmosphere (as a function of absorbed radiation and vapour pressure deficit) which results in robust dynamics in response to varied water availability (Bonan et al., 2014). We create a very large ensemble of SPA runs across environmental space to map the sensitivity of GPP and ET to biophysical changes, and then fit the parameters of ACM-GPP-ET to theses surfaces.

     ACM-GPP-ET is based on previous approaches developed to estimate GPP and ET independently (Williams et al., 1997;

Fisher et al., 2008), but here uniquely are realistically coupled for the first time. GPP is estimated as a function of foliar nitrogen content allocated to photosynthetic activity, temperature, intercellular $CO_2$ concentration and absorbed PAR. ET is estimated as the sum of transpiration, evaporation from the soil surface and of rainfall intercepted by the canopy, within a soil water mass balanced system. Using a combination of GPP and ET estimates from both TEM and observation-orientated analyses spanning site to global scales we calibrate and validate ACM-GPP-ET and address the following questions:

1) How computationally efficient is ACM-GPP-ET compared to our complex TEM at estimating daily fluxes?

     2) How well can the intermediate complexity ACM-GPP-ET emulate the complex TEM (i.e. GPP, ET, their coupling via WUE and soil moisture)?

     3) How do ACM-GPP-ET predictions compare to fully independent FLUXNET derived estimates of carbon and water fluxes across the globe?

Finally we discuss novel research applications made possible using our intermediate complexity, ecophysiologically-based modelling approach including full Monte Carlo based quantification of model parameter and structural uncertainties, global scale sensitivity analyses (e.g. WUE response to increased $CO_2$), rapid testing of alternate theoretical models of stomatal conductance, and use within terrestrial ecosystem model-data fusion frameworks.

## 2 Description of ACM-GPP-ET

### 2.1 Model Overview

The Aggregated Canopy Model for Gross Primary Productivity and Evapo-Transpiration version 1 (ACM-GPP-ET v1) provides a computationally efficient yet broadly mechanistic representation of photosynthetic and evaporative fluxes of terrestrial ecosystems. Evapo-transpiration is explicitly represented as the sum of transpiration (coupled to GPP via a mechanistic representation of stomatal conductance), evaporation from the soil surface and evaporation of precipitation intercepted by the canopy. Absorption and reflectance of short- and long-wave radiation are estimated as non-linear functions of LAI. Aerodynamic conductance for canopy and soil surface exchange are estimated as a function of wind speed and canopy structure (LAI and height). ACM-GPP-ET includes a four-layer model of soil water balance. The top three soil layers are accessible to roots which determines the available supply of water to the plant as a function of fine root biomass and their distribution through the soil profile. Soil evaporation is assumed to be supported by the top soil layer only (Figure 1).

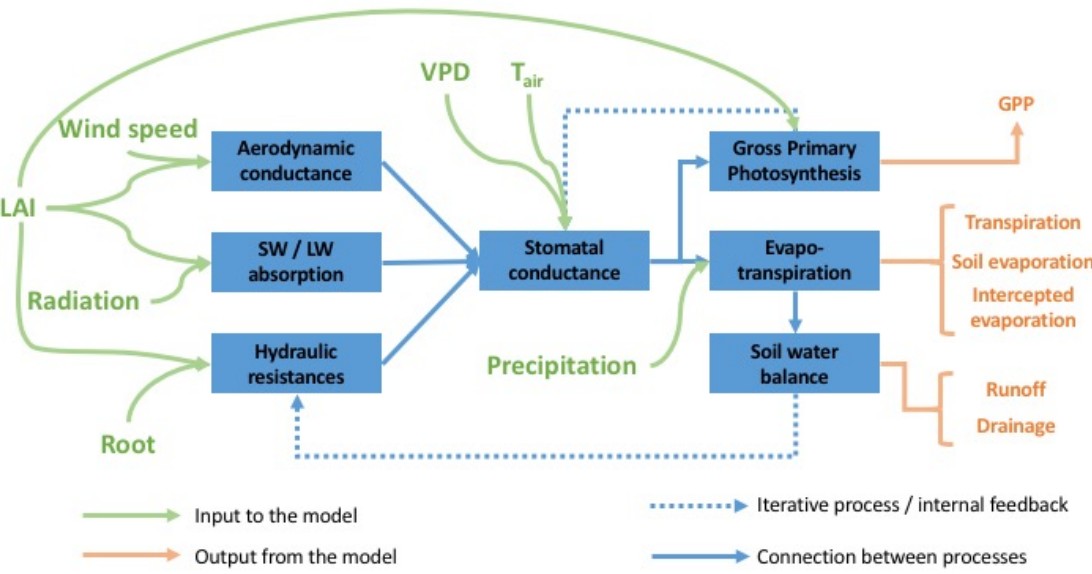

**Figure 1.** Schematic of ACM-GPP-ET showing its inputs, outputs and how its processes are interconnect. The blue boxes indicate distinct process groupings within the model framework. Green arrows are inputs to the model, while orange arrows indicate model output (i.e. carbon / water fluxes). Blue arrows show the interconnections between the various processes.

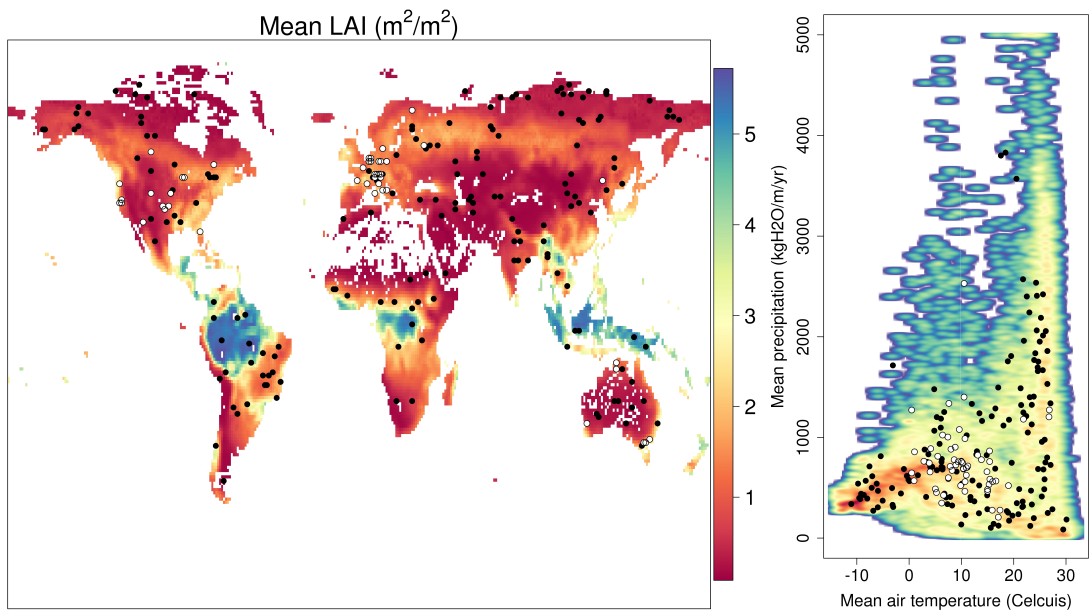

**Figure 2.** Left global (1 x 1 degree) mean LAI estimates (2001-2015) derived from NASA's MODIS product. Right climate space described by mean annual precipitation and temperature (ECMWF's ERA-Interim; Dee et al., 2011). Shading indicates the point density with red areas indicating high density. In both plots black circles show the location of SPA calibration sites, while the white circles with black ring show the FLUXNET2015 sites used to provide independent validation.

## 2.2 Model Drivers

ACM-GPP-ET requires both meteorological and biophysical information as inputs (Table 1). Most of the needed drivers are widely available from either field observations or global re-analyses. Meteorological drivers are extracted from the European Centre for Medium Range Weather Forecasting (ECMWF) ERA-Interim re-analysis (Dee et al., 2011), while soil textural

5   information are extracted from global interpolations of field inventories (e.g., HWSD; Hiederer and Köchy, 2011). LAI is widely available from satellite based remote sensing such as the NASA generated MODIS product (https://modis.gsfc.nasa.gov/data/dataprod/mod15.php). In contrast, information on below ground biophysical information, such as root stocks and rooting depth, is more challenging to obtain as they are highly spatially variable and not directly observable from space. Simulation models and model-data fusion based C-cycle analyses may simulate root stocks which can provide useful information (e.g.,

10  Bloom et al., 2016) while rooting depth information can also be statistically estimated (e.g., Fan et al., 2017).

**Table 1.** Drivers required as inputs by ACM-GPP-ET. Each driver has its unit specified and brief description. The mean value across the calibration climate space is given with the standard deviation in parenthesis. The drivers are divided between those which are time varying and those assumed constant for a given location but can vary between locations.

| | Name | Units | Mean (SD) | Description |
|---|---|---|---|---|
| **Time varying** | | | | |
| | Doy | Julian Day | - | Time step median Julian day of year |
| | Maximum temperature | °C | 14.9 (16.9) | Time step maximum air temperature |
| | Minimum temperature | °C | 7.3 (15.9) | Time step minimum air temperature |
| | Short-wave Radiation | MJ/m$^2$/day | 15.0 (8.4) | Time step average daily sum incoming short-wave radiation |
| | CO$_2$ | ppm or $\mu$mol/mol | 372.5 (42.7) | Time step mean atmospheric CO$_2$ concentration |
| | VPD | Pa | 753 (888) | Time step mean water vapour pressure deficit |
| | Precipitation | kgH$_2$O/m$^2$/s | 2.7x10$^{-6}$ (9.6x10$^{-6}$) | Time step mean liquid precipitation rate |
| | Wind Speed | m/s | 2.9 (1.8) | Time step mean wind speed |
| | LAI | m$^2$/m$^2$ | 1.9 (1.3) | Time step specific leaf area index |
| | Fine root stocks | gC/m$^2$ | 151 (105) | Time step specific fine root stocks |
| **Constant** | | | | |
| | $D_{root_{max}}$ | m | 2 | Maximum rooting depth |
| | $K_{root}$ | gbiomass/m$^2$ | 150 | Root biomass needed to reach 50 % of $D_{root_{max}}$ |
| | Soil sand percentage | volumetric percentage | 45.8 (16) | Soil sand percentage representative of two depths, 0-30 cm and 31-100 cm |
| | Soil clay percentage | volumetric percentage | 21.4 (8.8) | Soil clay percentage representative of two depths, 0-30 cm and 31-100 cm |

## 2.3 Model Parameters

Parameters within ACM-GPP-ET represent a wide range of time-invariant physical, biogeophysical and biogeochemical properties. In this study we calibrate a total of 22 parameters related to nitrogen-use and light-use efficiency, temperature response of photosynthesis, plant water-use efficiency and radiation absorption and reflectance processes (Table 2); these parameters broadly relate to ecosystem traits. Ecosystem traits can be reasonably expected to vary between ecosystems and thus we should be able to retrieve ecologically consistent estimates for these parameters given suitable carbon and water flux information. In this study were we are calibrating against a complex model with 'known' parameter values against which we can compare our estimated values. There are a further 12 biophysical parameters which are assumed to be constant and therefore are not retrieved as part of our calibration procedure (Table 3).

## 2.4 Gross Primary Productivity

Following Williams et al. (1997) and Jones (1992), GPP is estimated as a co-limited function of temperature, CO$_2$ (limited by stomatal opening and thus plant water availability) and absorbed PAR. The temperature and foliar nitrogen limited rate of photosynthesis (P$_{NT}$; gC/m$^2$/day) is first determined as a function of leaf area index (LAI; m$^2$/m$^2$), average foliage nitrogen content (N$_{fol}$; gN/m$^2$leaf), nitrogen use efficiency (NUE; gC/gN/m$^2$leaf/day) and temperature (T$_{air}$; °C).

$$P_{NT} = \text{LAI} \cdot N_{fol} \cdot \text{NUE} \cdot T_{adj} \tag{1}$$

**Table 2.** ACM-GPP-ET parameters retrieved using the CARDAMOM model-data fusion system. For each parameter its symbol as used in the code description is given, along with the maximum and minimum values used for the prior ranges, the maximum likelihood estimate retrieved from the posterior distributions, units and a brief description. The SPA column contains the SPA parameter value used in the calibration where a direct equivalent is available. Near infrared radiation is abbreviated as NIR, photosynthetically active radiation is abbreviated as PAR and long-wave radiation as LW.

| Symbol | Prior (min/max) | Posterior | SPA | Units | Description |
|---|---|---|---|---|---|
| NUE | 3/40 | 14.9 | - | gC/gN/m$^2$leaf/day | Photosynthetic nitrogen use efficiency at optimum temperature, light and $CO_2$ conditions |
| $T_{max}$ | 45/60 | 52.6 | 56 | °C | Maximum temperature for photosynthesis |
| $T_{opt}$ | 20/40 | 34.5 | 30 | °C | Optimum temperature for photosynthesis |
| $Kurt$ | 0.1/0.3 | 0.13 | 0.183 | - | Kurtosis of temperature response |
| $E_0$ | 1/7 | 4.5 | | gC/MJ/m$^2$/day | Quantum yield, C update per unit of photosynthetically active radiation |
| $min\Psi$ | 2.5/1.5 | 2.0 | 2.0 | MPa | Absolute value for minimum tolerated leaf water potential |
| iWUE | 1x10$^{-10}$/1x10$^{-3}$ | 1.6x10$^{-6}$ | 8.8x10$^{-7}$ | gC/m$^2$leaf/day/mmolH$_2$O | Intrinsic water use efficiency |
| $\alpha_{LW-refl}$ | 0.9/1.0 | 0.07 | | - | Maximum fraction of incoming LW radiation reflectance by the canopy |
| $K_{LW-refl}$ | 0.01/2.5 | 0.79 | | m$^2$/m$^2$ | LAI at which LW radiation reflectance at 50 % of maximum |
| $\alpha_{NIR-refl}$ | 0.1/1.0 | 0.11 | | - | Maximum fraction of incoming NIR reflectance by canopy |
| $K_{NIR-refl}$ | 0.01/2.5 | 0.19 | | m$^2$/m$^2$ | LAI at which near NIR reflectance by canopy at 50 % of maximum |
| $\alpha_{PAR-refl}$ | 0.1/1.0 | 0.10 | | - | Maximum fraction of incoming PAR reflectance by canopy |
| $K_{PAR-refl}$ | 0.01/2.5 | 0.23 | | m$^2$/m$^2$ | LAI at which PAR reflectance by canopy at 50 % of maximum |
| $\alpha_{LW-trans}$ | 0.5/1.0 | 0.60 | | - | Maximum fractional reduction of incoming LW radiation transmitted by the canopy |
| $K_{LW-trans}$ | 0.01/2.5 | 0.51 | | m$^2$/m$^2$ | LAI at which reduction of LW radiation transmittance is at 50 % of maximum |
| $\alpha_{NIR-trans}$ | 0.5/1.0 | 0.99 | | - | Maximum fractional reduction of incoming NIR transmitted by canopy |
| $K_{NIR-trans}$ | 0.01/2.5 | 1.85 | | m$^2$/m$^2$ | LAI at which reduction of near NIR transmittance is at 50 % of maximum |
| $\alpha_{PAR-trans}$ | 0.5/1.0 | 0.99 | | - | Maximum fractional reduction of incoming PAR transmitted by canopy |
| $K_{PAR-trans}$ | 0.01/2.5 | 1.76 | | m$^2$/m$^2$ | LAI at which reduction of PAR transmittance is at 50 % of maximum |
| $\alpha_{LW-release}$ | 0.01/1.0 | 0.98 | | - | Maximum fraction of LW radiation emitted by leaf area to be released from the canopy |
| $K_{LW-release}$ | 0.01/2.5 | 0.68 | | m$^2$/m$^2$ | LAI at which LW release from the canopy is at 50 % of maximum |
| $soil_{abs}$ | 0.5/0.99 | 0.62 | 0.98 | - | Fraction of incident NIR + PAR absorbed by soil |

Where $T_{adj}$ describes a skewed normal distribution (scaling 0-1) with an optimum ($T_{opt}$), maximum temperature ($T_{max}$) and kurtosis ($Kurt$).

$$T_{adj} = exp\left( log\left( \frac{T_{max} - T_{air}}{T_{max} - T_{opt}} \right) \cdot Kurt \cdot (T_{max} - T_{opt}) \right) \cdot exp(Kurt \cdot (T_{max} - T_{opt})) \tag{2}$$

$CO_2$ limitation is depended on canopy conductance of $CO_2$ ($g_c$; mmolCO$_2$/m$^2$/day) which is assumed to be the combined conductance of the stomata ($g_s$; mmolH$_2$O/m$^2$/s) and leaf boundary layer ($g_{b-mmol}$; mmolH$_2$O/m$^2$/s). Note, that $g_s$ and $g_b$ are calculated for conductance of water vapour and thus coefficients 1.65 and 1.37 convert conductance of water to those of $CO_2$ (Jones, 1992), $dayl_{sec}$ = 86400 is the number of seconds per day.

$$g_c = dayl_{sec} \cdot \left( \frac{1}{g_{s-mmol} \cdot 1.65} + \frac{1}{g_{b-mmol} \cdot 1.37} \right)^{-1} \tag{3}$$

**Table 3.** Parameters describing physical or biophysical constants not retrieved as part of the CARDAMOM model-data fusion analysis.

| Symbol | Value | Units | Description |
|---|---|---|---|
| $PAR:SW$ | 0.5 | - | Fraction of short-wave radiation assumed to be photosynthetically active |
| $\sigma$ | 0.96 | - | Long-wave radiation emissivity of a surface |
| $\kappa$ | $5.67 \times 10^{-8}$ | W/m$^2$/K$^{-4}$ | Stefan-Boltzmann constant |
| $d_o$ | 0.08 | m | Mean leaf diameter |
| $Root_{density}$ | $0.31 \times 10^6$ | gbiomass/m$^3$ | Mean root density within soil per unit volume (Bonan et al., 2014) |
| $Root_{radius}$ | 0.00029 | m | Mean root radius (Bonan et al., 2014) |
| $Root_{Resist}$ | 25 | MPa/s/g/mmolH$_2$O | Mean root resistivity to hydraulic flow (Bonan et al., 2014) |
| $G_p$ | 5 | mmol/m$^2$leaf/s/MPa | Mean plant conductivity to hydraulic flow (Bonan et al., 2014) |
| $\alpha_{comp}$ | 36.5 | $\mu$mol/mol CO$_2$ | CO$_2$ compensation point for photosynthesis at 20$^o$C (McMurtie et al., 1992) |
| $\alpha_{half}$ | 310 | $\mu$mol/mol CO$_2$ | CO$_2$ half saturation point for photosynthesis at 20$^o$C (McMurtie et al., 1992) |
| $\beta_{comp}$ | 282.61 | K | Temperature sensitivity parameter for CO$_2$ compensation point (McMurtie et al., 1992) |
| $\beta_{half}$ | 297.106 | K | Temperature sensitivity parameter for CO$_2$ half saturation point (McMurtie et al., 1992) |

The canopy boundary layer ($g_b$; see Eq. 57) and stomatal conductance ($g_s$) are initially calculated in m/s, and thus must be converted into mmolH$_2$O/m$^2$/s for the purposes of calculating CO$_2$ exchange. Note that the coupling between photosynthesis and transpiration occurs via $g_s$ which is estimated as a function of available water supply, atmospheric demand and the intrinsic water use efficiency threshold on GPP, see Sec. 2.6 for details on the estimation of $g_s$.

$$g_{b-mmol} = g_b \cdot (1000 \cdot \text{Pr}_{air}/(\text{T}_{airK} \cdot \text{R}_{con})) \tag{4}$$

$$g_{s-mmol} = g_s \cdot (1000 \cdot \text{Pr}_{air}/(\text{T}_{airK} \cdot \text{R}_{con})) \tag{5}$$

Where $\text{T}_{airK}$ is the air temperature in Kelvin, $\text{Pr}_{air}$ is air pressure (default = 101325 Pa) and $\text{R}_{con}$ is the universal gas constant (8.3144 J/K/mol). The scalar 1000 adjusts units from mol to mmol.

The internal CO$_2$ concentration ($\text{C}_i$; ppm or $\mu$mol/mol) is estimated as a function of atmospheric CO$_2$ concentration ($\text{C}_a$; ppm or $\mu$mol/mol), $g_c$, the CO$_2$ compensation ($\text{C}_{comp}$; ppm or $\mu$mol/mol) and half saturation ($\text{C}_{half}$; ppm or $\mu$mol/mol) points.

$$\text{C}_i = \frac{m + (m^2 - 4 \cdot (\text{C}_a \cdot q - p \cdot \text{C}_{comp}))^{0.5}}{2} \tag{6}$$

$$q = \text{C}_{comp} - \text{C}_{half} \tag{7}$$

$$p = (\text{P}_{NT} \cdot \text{M}_{\text{C}}^{-1} \cdot 1\text{x}10^6)/g_c \tag{8}$$

$$m = \text{C}_a + q - p \tag{9}$$

$\text{M}_{\text{C}}$ (12 gC/mol) is the molar ratio of carbon, where its inverse ($\text{M}_{\text{C}}^{-1}$ converts from molC to gC and $1\text{x}10^6$ scales from $\mu$mol to mol.

$\text{C}_{comp}$ determines the $\text{C}_i$ at which GPP becomes positive while $\text{C}_{half}$ is the $\text{C}_i$ at which CO$_2$ limited photosynthesis is at 50 % of its maximum rate. Both $\text{C}_{comp}$ and $\text{C}_{half}$ are calculated as a function of temperature following McMurtie et al. (1992).

$$\text{C}_{comp} = \alpha_{comp} \cdot e^{\beta_{comp} \cdot \frac{\text{T}_{airK} - 298.15}{\text{T}_{airK}}} \tag{10}$$

$$\text{C}_{half} = \alpha_{half} \cdot e^{\beta_{half} \cdot \frac{\text{T}_{airK} - 298.15}{\text{T}_{airK}}} \tag{11}$$

where $\alpha_{comp}$ and $\alpha_{half}$ describe the values at the reference temperature (20$^o$C or 298.15 K) and $\beta_{comp}$ and $\beta_{half}$ describe the sensitivity of the temperature response.

$CO_2$ limited photosynthesis ($P_{CO2}$; gC m$^{-2}$ day$^{-1}$) is calculated as a function of $g_c$ and $CO_2$ exchange gradient. Where 1x10$^{-6}$ scales from $\mu$mol to mol while $M_C$ molar units to gC. At this juncture, a day length (dayl: hours) correction is applied to be consistent with the light limitation calculation which follows

$$P_{CO2} = (g_c \cdot (C_a - C_i)) \cdot 1x10^{-6} \cdot M_C \cdot \frac{dayl}{24} \tag{12}$$

Light limited photosynthesis ($P_I$; gC/m$^2$/day) is defined as a function of absorbed short-wave radiation (I) and a quantum yield parameter ($E_0$).

$$P_I = E_0 \cdot I \tag{13}$$

The final GPP estimate (gC/m$^2$/day) is the result of combined light and $CO_2$ limited photosynthesis.

$$GPP = \frac{P_I \cdot P_{CO2}}{P_I + P_{CO2}} \tag{14}$$

## 2.5 Evapo-transpiration

Evapo-transpiration is based on the Penman-Monteith model assuming isothermal net radiation conditions (Jones, 1992). Evaporation is simulated from three source which are (i) transpiration, (ii) evaporation of precipitation intercepted by the canopy and (iii) the soil surface. The following sections detail the calculation of each evaporative source within their respective available water supplies.

### 2.5.1 Transpiration

Transpiration ($E_{trans}$; kgH$_2$O/m$^2$/day) is estimated by the Penman-Monteith equation linking the drivers of transpiration, canopy radiation status and atmospheric demand, with restrictions on evaporative losses, namely available water supply from the roots within the soil profile. The upper limit on water supply is imposed by restricting the maximum stomatal conductance ($g_s$) for a given set of environmental conditions (process described in Sect. 2.6).

$$E_{trans} = \frac{(s \cdot \Phi_{iso-canopy}) + (\rho_{air} \cdot cp_{air} \cdot VPD \cdot g_b)}{\lambda \cdot (s + (\gamma \cdot (1 + g_b/g_s)))} \cdot dayl \cdot 3600 \tag{15}$$

$\Phi_{iso-canopy}$ is the isothermal net radiation (W/m$^2$; see Section 2.7.2) while $s$ (kPa K$^{-1}$) is the slope of curve relating saturation vapour pressure with air temperature and $\gamma$ is the psychrometer constant (kPa K$^{-1}$). $\rho_{air}$ is the density of air (kg/m$^3$), $\lambda$ is the latent heat of vaporisation (J kg$^{-1}$) and $cp_{air}$ is the specific heat capacity of air (J/kg/K).

$s$, $\gamma$ and $\lambda$ are calculated as a function of $T_{air}$ following equations described in Jones (1992).

$$s = \frac{s_{ref0} \cdot 17.269 \cdot 237.3 \cdot e^{17.269 \cdot T_{air}/(T_{air}+237.3)}}{(T_{air} + 237.3)^2} \tag{16}$$

$$\gamma = \gamma_{ref0} \cdot e^{0.00097 \cdot T_{air}} \tag{17}$$

$$\lambda = \lambda_{ref0} - 2364 \cdot T_{air} \tag{18}$$

$s_{ref0}$ (0.61078 MPa), $\gamma_{ref0}$ (0.0646 kPa) and $\lambda_{ref0}$ (2501000 J/kg) are $s$, $\gamma$ and $\lambda$ at $0^oC$ reference temperature. $\rho_{air}$ is calculated as a function of $T_{airK}$.

$$\rho_{air} = \frac{353}{T_{airK}} \tag{19}$$

The calculation of canopy conductance ($g_b$; m/s) is described in section 2.8.1 linked to canopy properties (LAI and canopy height) and wind speed. Stomatal conductance ($g_s$; m/s), used in both the calculation of GPP and transpiration, is calculate via an iterative bisection procedure described in the following section.

### 2.5.2 Wet canopy evaporation

Wet canopy surface evaporation ($E_{wet}$; $kgH_20/m^2/day$) is the evaporation of precipitation intercepted by the canopy and thus is limited by the available canopy water storage ($C_{stor}$).

$$E_{wet} = \begin{cases} C_{stor}, & E_{pot} > C_{stor} \\ E_{pot}, & \text{otherwise} \end{cases} \tag{20}$$

Where $E_{pot}$ is the potential wet canopy evaporation ($kgH_20/m^2/day$). $E_{pot}$ is assumed to be unrestricted evaporation as estimated by the Penman model assuming isothermal net radiation. $E_{pot}$ is further restricted based on the ratio of current canopy water storage

$$E_{pot} = \left( \frac{(s \cdot \Phi_{iso-canopy}) + (\rho_{air} \cdot cp_{air} \cdot VPD \cdot g_b)}{\lambda \cdot (s + \gamma)} \cdot dayl_{sec} \cdot \frac{C_{stor}}{C_{max}} \right) \tag{21}$$

Where $C_{max}$ is the maximum canopy water storage ($kgH_20/m^2$), defined as a function of LAI related by $\alpha$ (0.2) as previously used in SPA (Smallman et al., 2013).

$$C_{max} = \alpha \cdot LAI \tag{22}$$

$C_{stor}$ is determine by water inputs from precipitation, less that which reaches the soil surface (i.e. through-fall), and its water losses by evaporation (as described above) or overflow from intercepted water exceeding $C_{max}$ onto the ground. The fraction of precipitation expected to be through-fall (Tfall) is estimated as a function of LAI related by $\mu$ (0.5). Where $\mu$ is selected assuming that interception if rainfall is similar to that of direct radiation.

$$Tfall = e^{-\mu \cdot LAI} \tag{23}$$

### 2.5.3 Soil surface evaporation

Soil evaporation ($E_{soil}$; $kgH_20/m^2/day$) is estimated using the Penman-Monteith equation linking drivers of evaporation, soil isothermal radiation ($\Phi_{iso-soil}$; $W/m^2$) and atmospheric demand, with restrictions on evaporative losses (i.e. namely available

water in the top soil layer). The upper limits evaporation is also restricted by the thickness of the dry layer (drythick; m) of soil at the surface.

$$\text{E}_{soil} = \frac{(s \cdot \Phi_{iso-soil}) + (\rho_{air} \cdot cp_{air} \cdot \text{VPD}_{soil} \cdot \text{g}_{soil})}{\lambda \cdot (s + (\gamma \cdot (1 + \text{g}_{soil}/\text{g}_{ws})))} \cdot \text{dayl} \cdot 3600 \tag{24}$$

Where $\text{g}_{soil}$ is the soil surface aerodynamic conductance (m/s) and $\text{g}_{ws}$ is the conductance of water vapour (m/s) through the soil air space. $\text{VPD}_{soil}$ is the vapour pressure deficit (kPa) between the air above the soil and air within the soil pore space.

$$\text{g}_{ws} = \frac{\text{por}_{top} \cdot D_w \cdot \left(\frac{\text{T}_{airK}}{293.2}\right)^{1.75}}{\tau \cdot \text{drythick}} \tag{25}$$

Where $\text{por}_{top}$ is the porosity (0-1; m$^3$/m$^3$) for the top soil layer as calculated using the Saxton model of soil hydrology (Saxton et al., 1986; Williams et al., 2001b), $D_w$ is the diffusion coefficient for water through air (m$^2$/s) at reference temperature 293.2 K and 1.75 is a scalar coefficient relating the temperature dependence of $D_w$. $\tau$ is the tortuosity (=2.5).

$$\text{VPD}_{soil} = \text{VPD} - e_{surf} \tag{26}$$

$$e_{surf} = e_{sat} - e_{soil} \tag{27}$$

$$e_{soil} = e_{sat} \cdot e^{\frac{1\text{x}10^6 \cdot \text{SWP}_{top} \cdot V_w}{R_{con} \cdot \text{T}_{airK}}} \tag{28}$$

$$e_{sat} = 0.1 \cdot e^{\frac{1.8095 + (17.269 \cdot \text{T}_{airK} - 4717.306)}{\text{T}_{airK} - 35.86}} \tag{29}$$

$e_{surf}$ is the vapour pressure deficit within the soil air space (kPa). $e_{soil}$ is the vapour pressure in the soil air space (kPa) and $e_{sat}$ is the saturation vapour pressure at the current temperature. $\text{SWP}_{top}$ is the soil water potential (MPa) for the top soil layer while $V_w$ is the partial molar volume of water at 20 °C (= 1.805x10$^{-7}$ m$^3$/mol). All other scalar values are coefficients relating current air temperature to $e_{sat}$.

## 2.6 Calculating g$_s$: the coupling point between plant C and H$_2$O cycles

The iWUE optimisation approach for estimating g$_s$ is well established and validated, in particular iWUE has been show to show improved drought response of g$_s$ compared to Ball-Berry g$_s$ model Williams et al. (1996); Bonan et al. (2014). The model aims to maximise photosynthetic uptake within the constraints on g$_s$ imposed by the available supply of water to the canopy and atmospheric demand for evaporation, this approach is referred to as optimising the intrinsic water use efficiency (iWUE).

Calculation of g$_s$ is a three step process. Step 1 estimation of the potential steady flow water supply over the day (MaxSupply; kgH$_2$0/m$^2$/day) from the soil via roots to the canopy.

$$\text{MaxSupply} = \frac{LWP_{min} - \text{wSWP}}{\text{R}_{tot}} \cdot \text{M}_{\text{H2O}} \cdot 1\text{x}10^{-3} \cdot dayl_{sec} \tag{30}$$

Where $LWP_{min}$ is the minimum tolerated leaf water potential (MPa), wSWP is the soil water potential weighted by root access (MPa) and $R_{tot}$ is the total hydraulic resistance (MPa/s/m$^2$/mmol). The unit is changed from mmol to gC using the molar mass of water $M_{H2O}$ (18 g/mol) and 1x10$^{-3}$ scalar. Step 2 inverts the Penman-Monteith equation to calculate the value of $g_s$ required to meet MaxSupply under current atmospheric demand and isothermal net radiation conditions.

$$g_{s-max} = g_b / \frac{\left( \frac{(s \cdot \Phi_{iso-canopy}) + (\rho_{air} \cdot cp_{air} \cdot \text{VPD} \cdot g_b)}{(\lambda \cdot (\text{MaxSupply}/(dayl_{sec} \cdot \frac{dayl}{24}))) - s} \right)}{\gamma} \tag{31}$$

Step 3 uses an iterative bisection process which quantifies the sensitivity of GPP to $g_s$ increment by 1 mmolH$_2$0/m$^2$/s ($\delta$GPP; gC/m$^2$leaf/day/mmolH$_2$0g$_s$); between $g_s = 0$ and $g_{s-max}$ minimising gs$_{opt}$, the difference between $\delta$GPP/g$_s$ and $iWUE$; gC/m$^2$leaf/day/mmolH$_2$0g$_s$.

$$\text{gs}_{opt} = \text{iWUE} - \frac{\delta\text{GPP}}{\text{LAI}} \tag{32}$$

## 2.7  Radiation balance

State-of-the-art radiative transfer schemes are able to quantify differential canopy absorption, transmittance to soil soil surface and reflection back to the sky of PAR, NIR and long-wave radiation. Using a detailed radiative transfer scheme as a base (Williams et al., 1998), here we have developed simple Michaelis-Menten relationships parameterised to reproduce the emergent absorption, transmittance and reflection properties of a canopy as a function of LAI.

Net canopy ($\Phi_{iso-canopy}$; W/m$^2$) and soil ($\Phi_{iso-soil}$; W/m$^2$) isothermal radiation balances are calculated from the combination of short- and long-wave absorption detailed in the following sections.

$$\Phi_{iso-canopy} = APAR_{canopy} + ANIR_{canopy} + ALW_{canopy} \tag{33}$$

$$\Phi_{iso-soil} = APAR_{soil} + ANIR_{soil} + ALW_{soil} \tag{34}$$

### 2.7.1  Short-wave radiation absorption

ACM-GPP-ET uses a bi-directional radiative transfer scheme to estimate the absorption of PAR and NIR by the canopy and soil surface. Downward radiation first interacts with the canopy either being reflected back toward the sky, transmitted toward the soil surface or absorbed by the canopy. Second the radiation which is transmitted through the canopy to the soil surface is either absorbed or reflected back through the canopy.

The fraction of incoming PAR ($canopy_{PAR-abs}$) and NIR ($canopy_{NIR-abs}$) absorbed is estimated as the residual of that reflected back into the sky ($canopy_{PAR-refl}$, $canopy_{NIR-refl}$) or that transmitted ($canopy_{PAR-trans}$, $canopy_{NIR-trans}$)

toward the soil surface.

$$canopy_{PAR-refl} = \frac{\alpha_{PAR-refl} \cdot \text{LAI}}{\text{LAI} + K_{PAR-refl}} \tag{35}$$

$$canopy_{NIR-refl} = \frac{\alpha_{NIR-refl} \cdot \text{LAI}}{\text{LAI} + K_{NIR-refl}} \tag{36}$$

$$canopy_{PAR-trans} = 1 - \frac{\alpha_{PAR-trans} \cdot \text{LAI}}{\text{LAI} + K_{PAR-trans}} \tag{37}$$

$$canopy_{NIR-trans} = 1 - \frac{\alpha_{NIR-trans} \cdot \text{LAI}}{\text{LAI} + K_{NIR-trans}} \tag{38}$$

$$canopy_{PAR-abs} = 1 - canopy_{PAR-refl} - canopy_{PAR-trans} \tag{39}$$

$$canopy_{NIR-abs} = 1 - canopy_{NIR-refl} - canopy_{NIR-trans} \tag{40}$$

where $\alpha_{NIR-refl}$ and $\alpha_{PAR-refl}$ are the maximum fraction of NIR and PAR reflected by the canopy. $K_{NIR-refl}$ and $K_{PAR-refl}$ are the LAI values at which 50 % of maximum reflectance is achieved for NIR and PAR respectively. $\alpha_{NIR-trans}$ and $\alpha_{PAR-trans}$ are the maximum reduction in transmittance for NIR and PAR, similarly $K_{NIR-trans}$ and $K_{PAR-trans}$ are the LAI at which transmittance is reduced by 50 %. Absorption of PAR ($APAR_{canopy}$) and NIR ($ANIR_{canopy}$) by the canopy on its first pass down through the canopy is estimate as

$$APAR_{canopy} = \text{PAR} \cdot canopy_{PAR-abs} \tag{41}$$

$$ANIR_{canopy} = \text{NIR} \cdot canopy_{NIR-abs} \tag{42}$$

Transmitted PAR and NIR is then incident on the soil surface to be absorbed by the soil surface or reflected back up towards the canopy. We assume that the soil absorption fraction ($soil_{abs}$) of incident PAR and NIR are the same, however PAR and NIR remain independently tracked to allow for subsequent reflection back towards the canopy.

$$APAR_{soil} = \text{PAR} \cdot canopy_{PAR-trans} \cdot soil_{abs} \tag{43}$$

$$ANIR_{soil} = \text{NIR} \cdot canopy_{NIR-trans} \cdot soil_{abs} \tag{44}$$

PAR and NIR which is reflected from the soil is then available for a second opportunity of the canopy to absorb, and typically contributed < 1 % of absorbed radiation in ACM-GPP-ET. Therefore the total $APAR_{canopy}$ and $ANIR_{canopy}$ are calculated as follows

$$APAR_{canopy} = APAR_{canopy} + (\text{PAR} \cdot canopy_{PAR-trans} \cdot (1 - soil_{abs}) \cdot canopy_{PAR-abs}) \tag{45}$$

$$ANIR_{canopy} = ANIR_{canopy} + (\text{NIR} \cdot canopy_{NIR-trans} \cdot (1 - soil_{abs}) \cdot canopy_{NIR-abs}) \tag{46}$$

Estimates of incoming short-wave radiation are widely available, however partitioned estimates of NIR and PAR are less frequent. In such circumstances we assume a fixed ratio between PAR and total short-wave radiation ($PAR : SW$).

$$\text{PAR} = \text{SW} \cdot PAR : SW \tag{47}$$

$$\text{NIR} = \text{SW} - \text{PAR} \tag{48}$$

### 2.7.2 Isothermal long-wave radiation absorption

The long-wave radiation balance is estimated assuming isothermal conditions (i.e. the surfaces are assumed to be the same temperature as the surrounding air). Calculation of the isothermal long-wave radiation is a trade-off between need to account for a positive bias in available energy for evaporation should only the short-wave radiation be accounted for, and errors introduced by not considering thermal heating and cooling of surfaces.

Similar to the short-wave radiative transfer scheme described above ACM-GPP-ET uses a bi-directional radiative transfer scheme the estimate the absorption and emission of long-wave by the canopy and soil surface. The long-wave radiation balance is divided into four components. First, incoming radiation interacts with the canopy either being reflected back toward the sky, transmitted toward the soil surface or absorbed by the canopy. Second, the radiation which is transmitted through the canopy to the soil surface is either absorbed or reflected back through the canopy. Third, the soil surface emits long-wave radiation towards the canopy, repeating step 1 in reverse. Fourth, the canopy itself emits long-wave radiation which is either incident on the soil surface or lost to the sky.

Emission of long-wave radiation (W/m$^2$) is dependent on temperature and emissivity ($\sigma$) related by the Stefan-Boltzmann's constant ($\kappa$). Incoming long-wave radiation (LW) from the sky is assumed to be related to surface air temperature ($T_{airK}$) minus 20 °C, while long-wave emission (LW$_{em}$) from surfaces is assumed to be related to surface air temperature under isothermal conditions.

$$\text{LW} = \sigma \cdot \kappa \cdot (T_{airK} - 20)^4 \tag{49}$$

$$\text{LW}_{em} = \sigma \cdot \kappa \cdot (T_{airK})^4 \tag{50}$$

The fraction of incoming LW to be reflected ($canopy_{LW-refl}$), transmitted ($canopy_{LW-trans}$) and absorbed ($canopy_{LW-abs}$) by the canopy are estimates as a function of LAI.

$$canopy_{LW-refl} = \frac{\alpha_{LW-refl} \cdot \text{LAI}}{\text{LAI} + K_{LW-refl}} \tag{51}$$

$$canopy_{LW-trans} = 1 - \frac{\alpha_{LW-trans} \cdot \text{LAI}}{\text{LAI} + K_{LW-trans}} \tag{52}$$

$$canopy_{LW-abs} = 1 - canopy_{LW-refl} - canopy_{LW-trans} \tag{53}$$

The canopy emits long-wave radiation (=LW$_{em}$·LAI) much of which it absorbed within the canopy itself, resulting in a decreasing fraction of long-wave emitted by the canopy from actually leaving the canopy airspace ($canopy_{LW-release}$).

$$canopy_{LW-release} = 1 - \frac{\alpha_{LW-release} \cdot \text{LAI}}{\text{LAI} + K_{LW-release}} \tag{54}$$

The soil surface also emits long-wave radiation (=LW$_{em}$) which is absorbed, reflected or transmitted through the canopy above. Net absorption of long-wave radiation by the canopy is therefore calculated as

$$ALW_{canopy} = \text{LW} \cdot canopy_{LW-abs} + \text{LW}_{em} \cdot canopy_{LW-abs} - \text{LW}_{em} \cdot \text{LAI} \cdot 2 \cdot canopy_{LW-release} \tag{55}$$

Note the factor 2 refers to the two sides of a leaf both of which are releasing long-wave radiation, one heading upwards towards the sky and the other heading downwards towards the soil. The net absorption of long-wave radiation by the soil is estimated as

$$ALW_{soil} = \sigma \cdot (\text{LW} \cdot canopy_{LW-trans} + \text{LW}_{em} \cdot \text{LAI} \cdot canopy_{LW-release}) - \text{LW}_{em} \tag{56}$$

where absorption of incident long-wave radiation is assumed to be equal to $\sigma$. We note a small quantity of long-wave emitted from the soil will be reflected back to the surface, however this is typically < 1 % of the long-wave energy budget and is neglected here.

## 2.8 Aerodynamic conductance

### 2.8.1 Canopy aerodynamic conductance

Canopy conductance of water vapour ($g_b$; m/s) and $CO_2$ is estimated using the leaf-level boundary layer conductance model used in SPA (Nikolov et al., 1995; Smallman et al., 2013). We assume that exchange is dominate at the top of the canopy and that conductance should be linked to the canopy top wind speed ($u_h$; m/s). Note that the boundary layer conductance model allows simulation of both free and forced convection, although here we simulate only the forced convection due to the lack of an explicit simulation of the energy balance. A detailed description of conductance model is given in Nikolov et al. (1995).

$$g_b = \frac{D_{wv} S_h}{d_o} \cdot 0.5 \cdot \text{LAI} \tag{57}$$

where $g_b$ is the conductance for water vapour (m/s), $D_{wv}$ is the temperature dependent molecular diffusivity of water vapour (m$^2$/s), where $D_{wv20}$ (0.0000242) is $D_{wv}$ at a 20$^o$C (293.15 K) reference temperature. $S_h$ the Sherwood number and $d_o$ is the leaf diameter.

$$D_{wv} = D_{wv20} \cdot \left(\frac{T_{airK}}{293.15}\right)^{1.75} \tag{58}$$

$S_h$ is estimated as a fraction of the Nusselt number ($N_u$), while $N_u$ is a function of the Prandtl ($P_r = 0.72$) and Reynolds ($R_e$) numbers.

$$S_h = 0.962 \cdot N_u \tag{59}$$

$$N_u = 1.18 \cdot P_r^{1/3} \cdot R_e^{1/2} \tag{60}$$

$$R_e = \frac{d_o \cdot u_h}{v} \tag{61}$$

$$v = \frac{\mu}{\rho_{air}} \tag{62}$$

$$\mu = \left(\frac{T_{airK}^{1.5}}{T_{airK} + 120}\right) \cdot 1.4963 \text{x} 10^{-6} \tag{63}$$

where $v$ is the kinematic viscosity (m$^2$/s) and $\mu$ is the dynamic viscosity (kg/m$^2$/s).

Above canopy momentum decay follows the standard log law decay assuming neutral conditions (Garratt, 1992).

$$u_h = \frac{u_*}{\kappa} \cdot log\left(\frac{z_h - d}{z_0}\right) \tag{64}$$

where $\kappa$ is Von Karman constant (0.41), $d$ is the canopy zero plane displacement height (m), $z_0$ is the canopy roughness length (m) and $u_*$ is the friction velocity (m/s). $d$ and $z_0$ are calculated based on canopy structure (height $z_h$ and LAI) as described in Raupach (1994).

$$d = z_h \left[1 - \frac{1 - \exp(-(C_{d1}\text{LAI})^{0.5})}{(C_{d1}\text{LAI})^{0.5}}\right] \tag{65}$$

and

$$z_0 = \left(1 - \frac{d}{z_h}\right) \exp\left(-\kappa\frac{u_h}{u_*} - \Psi_h\right) z_h, \tag{66}$$

where $C_{d1}$ is an empirically fitted parameter (7.5) and $\Psi_h$ (0.193) corrects the roughness length for the the effect of the roughness sub-layer. $u_*$ is estimated as a function of LAI and $u_h$

$$u_* = u_h \cdot (C_s + C_r \cdot \text{LAI} \cdot 0.5)^{0.5} \tag{67}$$

where $C_s$ = 0.003 approximates the impact of substrate drag and $C_r$ = 0.3 corrects for the roughness sub-layer (Raupach, 1994).

### 2.8.2 Soil aerodynamic conductance

Soil aerodynamic conductance ($g_{\text{soil}}$; $\text{m s}^{-1}$) is first calculated as a resistance. Soil resistance is integrated from the soil roughness length ($z_{soil}$ = 0.001 m) through the canopy based on the turbulent eddy diffusivity following Niu and Yang (2004).

$$r_{\text{soil}} = \int_{z_{\text{soil}}}^{d+z_0} dz/K_h(z), \tag{68}$$

where $dz$ is the vertical step size (m) through the canopy and $K_h$ is the eddy diffusivity at $z$ position (m) within the canopy. Eddy diffusivity ($K_h$; $\text{m}^2\,\text{s}^{-1}$) is assumed to have an exponential decay through the canopy (as with momentum). Eddy diffusivity at the canopy top is estimated as specified in Kaimal and Finnigan (1994).

$$K_h(z_h) = \kappa u_*(z_h - d) \tag{69}$$

$K_h$ is decayed through the canopy as described below.

$$K_h(z) = K_h(z_h) \exp\left(-f\left(1 - z/z_h\right)\right) \tag{70}$$

$$f = (c_d z_h \text{LAI}/l_m)^{0.5} (\Phi_m)^{0.5} \tag{71}$$

The coefficient of momentum decay $f$ is dependent on $c_d$ the coefficient of drag for foliage (0.2), LAI, $l_m$ and soil surface Monin-Obukov similarity coefficient ($\Phi_m$). $\Phi_m$ is assumed to be = 1 describing neutral conditions Garratt (1992).

## 2.9 Plant hydraulic resistance

We use a mechanistic model of plant hydraulics to determine the maximum available water supply to the canopy from each of the three potential rooting layers ($E_{layer}$; mmolH$_2$0/m$^2$/s) under steady state flow (Jones, 1992). The advantage to using a mechanistic approach allows for the estimation of physiological properties which makes possible novel comparisons with field observations such as Poyatos et al. (2016). The model assumes that the canopy is at $LWP_{min}$ drawing from each of the three soil layers based on their layer specific SWP, canopy, root and soil hydraulic resistances (MPa/s/m$^2$/mmol).

$$E_{layer} = \frac{|LWP_{min} - \text{SWP}| + (\rho_{lw} \cdot g \cdot z_h)}{\text{R}_{soil} + \text{R}_{root} + \text{R}_{canopy}} \tag{72}$$

Where $\rho_{lw}$ is the density of liquid water (1000 kg/m$^3$) and $g$ is the acceleration of gravity (9.82 m/s$^2$). The hydraulic resistance due to the soil ($\text{R}_{soil}$), roots ($\text{R}_{root}$) and the combined resistance of the stem and branch ($\text{R}_{canopy}$) each have units of MPa/s/m$^2$/mmol.

$$\text{R}_{soil} = \frac{ln(r_s/Root_{radius})}{2 \cdot \pi \cdot l_R \cdot l_s \cdot G_s} \tag{73}$$

$$\text{R}_{root} = \frac{Root_{resist}}{\text{C}_{root} \cdot 2 \cdot l_s} \tag{74}$$

$$\text{R}_{canopy} = \frac{z_h}{G_p \cdot \text{LAI}} \tag{75}$$

$Root_{radius}$ is the mean root radius (0.00029 m Bonan et al., 2014) and $r_s$ is the mean distance between roots (m). $l_R$ is the root length (m) within the current soil layer and $l_s$ is the thickness of the current soil layer (m). $Root_{resist}$ is the root resistivity (25 MPa/s/g/mmolH$_2$0; Bonan et al., 2014), $G_p$ is the plant conductivity to water (5 mmolH$_2$O/m$_{leafarea}$/s/MPa; Bonan et al., 2014).

$$r_s = \frac{1}{(l_R \cdot \pi)^{0.5}} \tag{76}$$

The root length ($l_R$) within each of the three soil layers available for root access is a function of available root biomass within that layer (Root$_{layer}$; g/m$^2$). The total root biomass (Root$_{total}$; g/m$^2$) is distributed between the three soil layers assuming that 50 % of the biomass is in the top 25 % of the rooting profile. Where the current rooting depth ($D_{root-cur}$; m) is assumed to follow an exponentially decaying function.

$$I_R = \frac{\text{Root}_{layer}}{Root_{density} \cdot pi \cdot Root_{radius}^2} \tag{77}$$

$$D_{root_cur} = \frac{Root_{max} \cdot \text{Root}_{total}}{K_{root} + \text{Root}_{total}} \tag{78}$$

$G_s$ is the soil conductivity (m$^2$/s/MPa) which is calculated as a function of soil textural parameters derived from the Saxton model of soil hydraulics ($sax_{c1}$, $sax_{c2}$ and $sax_{c3}$) and volumetric water content ($\Theta$; m$^3$/m$^3$). For further details see Saxton et al. (1986); Williams et al. (2001b)

$$G_s = sax_{c1} \cdot e^{\frac{sax_{c2} + sax_{c3}}{\Theta}} \tag{79}$$

The ratio of $E_{layer}/\Sigma E_{layer}$ determines the proportional extraction of water from each soil layer ($Up_{frac}$) due to $E_{trans}$.

$$Up_{frac} = \frac{E_{layer}}{\Sigma E_{layer}} \tag{80}$$

For use else where in the model the soil layer specific SWP and hydraulic resistances are aggregated based on uptake potential from each soil layer to provide an apparent SWP and resistance, i.e. the weighted soil water potential (wSWP) and total hydraulic resistance ($R_{tot}$).

$$\text{wSWP} = \Sigma(\text{SWP} \cdot Up_{frac}) \tag{81}$$

$$R_{tot} = \frac{\Sigma|LWP_{min} - \text{wSWP}|}{\Sigma E_{layer}} \tag{82}$$

## 2.10 Soil water balance

The Saxton model of soil hydraulics is used as the basis for simulation of the soil water balance within ACM-GPP-ET (Saxton et al., 1986). The implementation is a simplified version of that used within the SPA model (Williams et al., 2001b; Smallman et al., 2013). A total of four soil layers are simulated by the model, three of these layers are available for root access depending on the amount of root currently available. The first soil-layer has a fixed depth of 10 cm from which soil surface evaporation is extracted, while the second-layer has a fixed depth of 20 cm (i.e. total depth of first and second soil layers is 30 cm). The third layer has a variable depth dependent on the penetration depth of the roots within this layer (i.e. root biomass), thus providing a potential advantage of increasing rooting depth to access water resources deeper within the soil. The fourth soil layer is defined by the maximum soil rooting depth ($D_{root_{max}}$; m). The soil water mass balance is updated through four stages briefly described below.

The soil water mass balance is updated in sequence dealing will (i) evaporative losses, (ii) gravitational drainage, (iii) infiltration of precipitation and (iv) adjustments to the soil layers based on changes in rooting depth. Evaporative losses from the soil surface are extracted solely from the top soil layer, while water losses due to transpiration are extracted based on the $Up_{frac}$ as determined based on the rooting distribution. The gravitational drainage and infiltration schemes are a simplified implementation of those used by the SPA model (Williams et al., 2001b). Gravitational drainage is then calculated based on the downward flow of water from soil layers currently above their field capacity to deeper layers and ultimately out of the bottom of the soil water column (i.e. drainage flux). Precipitation which reaches the soil surface infiltrates based on the available pore space (i.e. porosity) of the soil layers. As the minimum time period used for the model is daily we assume that the maximum available pore space can be utilised. Once all soil layers have filled all available pore space (i.e. the soil is saturated) all remaining precipitation is assumed to be lost from the system as run-off.

## 3 Calibration procedure

We used the Soil Plant Atmosphere (SPA, Williams et al., 1996; Smallman et al., 2013) model to generate a data set of photo-synthetic and evaporative fluxes for the calibration of ACM-GPP-ET. The calibration of ACM-GPP-ET was conducted using

the CARDAMOM model-data fusion framework (Bloom and Williams , 2015). SPA simulated a 12 year period (2001-2012) at an hourly time step for 200 locations selected using a stratified random process from across the global land surface (Figure 2); stratification was to ensure even coverage across the latitudinal gradient. The number of sites selected was a trade-off to ensure good spatial coverage of training data but to end with a calibration dataset comprised of ∼50,000 days to reduce computational

cost for the calibration process. Land cover areas covered by desert, rocky areas or dominated by C4 photosynthetic pathway vegetation, as specified in the ECMWF land cover map, were excluded from the sampling to avoid areas which do not have substantial photosynthetic activity and to reflect the fact that ACM-GPP-ET is designed to simulate the dominant C3 photo-synthetic pathway.   To isolate the impact of root biomass and vertical distribution on water supply from the role soil moisture status SPA's soil moisture is held at field capacity for the creation of the calibration dataset. However. part of the validation

process (See section 4 for details) SPA is re-run but this time allowing the soil moisture status to vary in response to inputs and losses to facilitate validation of ACM-GPP-ET's capacity to simulate the development of drought compared to our complex model.

For both the calibration and validation simulations SPA (and ACM-GPP-ET) require inputs of meteorological drivers, foliar nitrogen, soil textural information and timeseries of LAI and root biomass. Prescribing LAI and root biomass, opposed to

coupling ACM-GPP-ET to a C-cycle model, allows the isolation of photosynthetic driver sensitivity without complex C-cycle feedbacks. Meteorological drivers were taken from the ERA-Interim reanalysis (Dee et al., 2011), these drivers were downscaled to an hourly time step using a weather generator (https://github.com/GCEL/WeatherGenerator_v1). Atmospheric $CO_2$ concentration for each day was sampled from 300-450 ppm; the exaggerated $CO_2$ range is to ensure that influence of increasing $CO_2$ concentrations is contained within the calibration dataset. Note that prescribing LAI, root biomass and fixing

soil moisture at field capacity prevents propagation of unexpected / undesired responses to strongly increasing atmospheric $CO_2$ gradients, i.e. each day's GPP and ET are purely as a function of that day's conditions. Mean foliar nitrogen content was randomly sampled for each site (but held constant over time) from log10 normal distribution (mean = 1.89 gN/m$^2$; Kattge et al., 2011). Soil sand and clay contents were extracted from the Harmonized World Soils Database (HWSD; Hiederer and Köchy, 2011), locations for which the sand / clay content fall outside the parameterised bounds for the Saxton soil hydrological model

were also excluded (∼0.7 % of global land surface). Timeseries of LAI and root stocks used to driver SPA and ACM-GPP-ET were extracted from a global terrestrial carbon cycle mode-data fusion analysis (Bloom et al., 2016). We used LAI and fine roots datasets from Bloom et al. (2016) derived from MODIS LAI products, remotely sensed above ground biomass and ecological process knowledge (for details see Bloom and Williams , 2015).

In order to quantify the ability of the analysis framework to retrieve accurate ecophysiological trait information (e.g. optimum

temperature of photosynthesis) a single set of ecophysiological parameters are used to drive SPA. This nominal parameter set are a combination of the forest hydraulic parameters from Williams et al. (1996) and broad-leaf forest radiation reflectance parameters Smallman et al. (2013).

The hourly SPA simulations were aggregated to daily time step and then sub-sampled at a 2-weekly interval to reduce temporal auto-correlation. Further filtering was applied to remove days with zero GPP (i.e. winter) and days for which SPA

was unable to solve its energy balance closure to a cumulative absolute error, over sub-daily time steps, of less than 50 W/m$^2$ summed across each 24-hour period. Thus, after filtering some 42,658 simulation days available for calibration.

## 3.1 SPA

The Soil Plant Atmosphere (SPA; Williams et al., 1996; Smallman et al., 2013) model simulates a mechanistic representation of the terrestrial ecosystem, coupling plant carbon and water cycles through ecophysiological principles. SPA simulates up to 10 canopy layers simulating both sunlit and shaded leaf area; each being independently connected to water supply from the soil. Water accessibility from 20 soil layers is determined as a function of root penetration within each soil layer. SPA estimates the surface exchanges of heat, water and $CO_2$ within a mass and energy balanced framework. SPA has been extensively validated at range of spatial scales (leaf to landscape) and climate zones (tropical, temperate and Arctic) (Williams et al., 1998, 2001b; Fisher et al., 2006, 2007; Sus et al., 2010; Smallman et al., 2013, 2014; López-Blanco et al., 2018). A detailed description of SPA and its major developments can be found in Williams et al. (1996, 1998, 2001b, 2005); Sus et al. (2010); Smallman et al. (2013), however a brief description follows below.

Leaf level photosynthesis (Farquhar; Farquhar and von Caemmerer, 1982), transpiration (Penman-Monteith; Jones, 1992) and energy balance are coupled via a mechanistic model of stomatal conductance (Williams et al., 1996; Bonan et al., 2014). Stomatal opening is modulated to optimise photosynthetic uptake within constraints determined by (i) the intrinsic water use efficiency and (ii) balancing atmospheric demand with available water supply from the soil. SPA makes use of a detailed multi-canopy layer radiative transfer scheme estimating the absorption and reflectance of both short- and long-wave radiation for the sunlit and shaded leaf area (Williams et al., 1998). Aerodynamic exchange coefficients for water, heat and $CO_2$ are estimated accounting for above, within and under-canopy momentum decay, including stability corrections based on Monin-Obukov stability theorem (Smallman et al., 2013).

## 3.2 CARDAMOM

The CARbon DAta MOdel fraMework (CARDAMOM; Bloom and Williams , 2015) is a model-data fusion (MDF) framework. CARDAMOM uses a Bayesian approach within a Metropolis Hastings - Markov Chain Monte Carlo (MH-MCMC) algorithm to retrieve parameters for a given model as a function of observational constraints. Parameter priors are specified with a uniform probability distribution truncated between minimum and maximum values (Table 1). CARDAMOM was used to calibrate ACM-GPP-ET based on GPP, transpiration, soil evaporation and evaporation of intercepted rainfall provided by SPA.

For the purpose of calculating the cost function used within the MDF approach we assumed uncertainties consistent with those expected from Eddy covariance. The uncertainty of both GPP and evaporative fluxes are assumed to be 15 %, representing random error estimates typically found for Eddy covariance (Stoy et al., 2006; Mauder et al., 2013). The use of the SPA model with known input parameters allows for quantification of how accurately the equivalent retrieved parameters have been determined. Where there is a directly equivalent parameter between SPA and ACM-GPP-ET the SPA values are provided in Table 1.

## 4  Validation procedure

ACM-GPP-ET is validated over two phases using a range of evapo-transpiration and GPP estimates from 1) out-of-sample SPA simulations and 2) fully independent eddy covariance derived estimates from the FLUXNET2015 database.

### 4.1  SPA validation

Phase 1) tests the ability of ACM-GPP-ET to simulate the soil moisture dynamics and resulting feedbacks on plant photosynthesis and evaporation estimated by the state-of-the-art SPA model. To achieve this we compare out-of-sample SPA simulated estimates of GPP and evapotranspiration. SPA's soil moisture status was held constant at field capacity during calibration phase and substantially sub-sampled. Here SPA's soil moisture status was allowed to vary as a function of its inputs and outputs, these simulations were carried out at the same locations as those used for the calibration analysis but without any sub-sampling.

### 4.2  Independent validation: FLUXNET2015

Phase 2) quantifies ACM-GPP-ET's predictive skill using fully independent observations. Furthermore, we contrast ACM-GPP-ET's predictive capacity with that of SPA. To achieve this comparison, both ACM-GPP-ET and SPA are simulated carbon and water fluxes at FLUXNET 2015 sites. This is achieved by both ACM-GPP-ET and SPA simulating carbon and water fluxes at FLUXNET2015 sites. The SPA simulations use the same nominal set of ecophysiological parameters used in generating the calibration dataset. ACM-GPP-ET uses the maximum likelihood parameter set retrieved during the calibration procedure. The parameter set used here broadly represent a forest ecosystem.Therefore, we hypothesise that both SPA and ACM-GPP-ET will perform best at forest sites and less well at sites with different hydraulic traits.

Daily estimates of GPP and ET derived from Eddy covariance were extracted from the FLUXNET2015 database (http://fluxnet.fluxdata.org/, accessed 01/11/2016). The FLUXNET2015 database was filtered to include only sites which overlapped our simulation period (2001-2015) for a minimum of three years, to allow for inter-annual comparison, and not more than 20 % missing data. ACM-GPP-ET is designed to emulate the C3 photosynthetic pathway, therefore sites which are listed to be dominated by vegetation which use the C4 or CAM pathways were removed. We also removed estimates which do not carry the highest quality flags to avoid comparing our model against estimates generates using a statistical model of net exchange (i.e. we use only the non-gap filled observations). Eddy covariance estimates where energy balance non-closure is less than 85 % were also rejected. In total 497 site years encompassing 10 vegetation types across 59 sites were available for validation of GPP and ET. Meteorological drivers used to driver SPA and ACM-GPP-ET were extracted from the the local site level measurements. Missing data were gap-filled either assuming linear interpretation (for single missing hours) or extracted from bias corrected temporally down-scaled meteorological drivers as used in the calibration process. Finally, as with the earlier SPA simulations LAI and root biomass for each site are extracted from Bloom et al. (2016).

## 5 Results

We show that a single global calibration of ACM-GPP-ET can effectively reproduce the patterns of GPP and ET simulated by SPA. Importantly the predictions of WUE are consistent for both ACM-GPP-ET and SPA, so that the simplified model is able to capture the interactions between C and water cycling. We also describe an independent validation against FLUXNET data, across 59 sites.

### 5.1 SPA calibration / validation

A single global ACM-GPP-ET parameterisation simulates the calibration dataset with a high degree of skill when using the maximum-likelihood parameter sets retrieved from the calibration analysis (Figure 3). The dynamics of soil evaporation is best simulated by ACM-GPP-ET followed by GPP and transpiration, each achieving $R^2 \geq 0.91$. Evaporation of canopy intercepted precipitation achieved a lower $R^2$ at 0.81. All fluxes are largely unbiased (GPP bias = -0.2gC/m$^2$/day and evaporative fluxes magnitude $\leq$0.08 kgH$_2$O/m$^2$/day) with low RMSE (0.97 gC/m$^2$/day and $\leq$0.39 kgH$_2$O/m$^2$/day). However, we note a tendency to underestimate peaks in transpiration found in the SPA simulation (Figure 3). Evaporation of canopy intercepted precipitation is least well simulated by each metric used here; however this is expected given the sensitivity of this flux to the timing and intensity of precipitation events and canopy energy balance varying strongly at a sub-daily time scales which are not accounted for here. Transpiration (E$_{trans}$; 61 %) dominates the overall ACM-GPP-ET simulated evaporative budget followed by evaporation of canopy intercepted rainfall (E$_{wet}$; 34 %) and soil surface evaporation (E$_{soil}$; 5 %), broadly consistent with those simulated by SPA in the calibration dataset (67 %, 28 % and 5 % respectively). ACM-GPP-ET is 2200 times faster than SPA, where ACM-GPP-ET requires $\sim$0.000007 seconds per day and SPA 0.015 seconds per day. Overall, ACM-GPP-ET simulates the calibration dataset with substantial skill and a substantial reduction in computational requirements.

Simulation of carbon and water fluxes remains robust in the phase 1) validation where ACM-GPP-ET is compared against the out-of-sample SPA simulations were soil moisture content is dynamically simulated, i.e. not held at field capacity as in the calibration procedure (Figure 4). Similarly, partitioning between evaporative fluxes remains closely aligned between ACM-GPP-ET (E$_{trans}$ = 59 %, E$_{wet}$ = 35 %, E$_{soil}$ = 6 %) and SPA (E$_{trans}$ = 62 %, E$_{wet}$ = 35 %, E$_{soil}$ = 7 %). Only soil evaporation suffers a substantial reduction in the simulation of the variability of SPA's soil evaporation from $R^2$ = 0.96 to 0.59, however remaining unbiased and RMSE increasing by only 0.02 kgH$_2$O/m$^2$/day (Figure 4).

Ecosystem water use efficiency (WUE = GPP / E$_{trans}$) simulated by SPA is well predicted by ACM-GPP-ET ($R^2$ = 0.79, RMSE = 1.88 gC/kgH$_2$O, bias = 0.33 gC/kgH$_2$O). The consistency within simulations with dynamic water availability demonstrates resilience in ACM-GPP-ET's ability to represent the linkages between the plant carbon and water cycles, which is key when considering the impacts of climatic extremes such as the evolution of drought. This ability to simulate drought in a manor consistent with SPA is supported by the high quality simulation of soil moisture content ($R^2$ = 0.84, RMSE = 4.19 kgH$_2$O/m$^2$, bias = 1.17 kgH$_2$O/m$^2$; Figure 4).

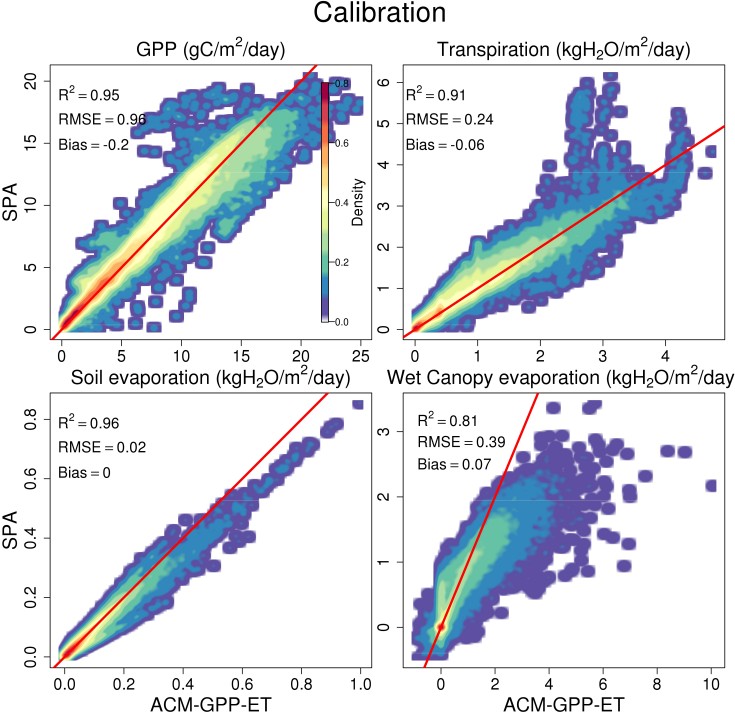

**Figure 3.** Comparison between SPA and ACM-GPP-ET generated fluxes, where ACM-GPP-ET used the maximum likelihood parameter sets for the calibration procedure. Fluxes compared are gross primary productivity (gC/m$^2$/day), transpiration (kgH$_2$O/m$^2$/day), soil evaporation (kgH$_2$O/m$^2$/day) and evaporation of water intercepted by the canopy (kgH$_2$O/m$^2$/day). The red line is a 1:1 line for reference. The colour intensity from blue to red indicates the density of flux estimates within a given area, note that for clarify showing low density areas the density values are scaled by density$^{1/4}$.

**Table 4.** Summary statistics for the global comparison between GPP and ET estimated by both ACM-GPP-ET, SPA and the FLUXNET2015 database. Statistics include R$^2$, root mean square error (RMSE) and mean bias (model-obs), GPP units are gC/m$_2$/day and ET are in units of kgH$_2$O/m$^2$/day.

| Model | Flux | R$^2$ | RMSE | Bias |
|---|---|---|---|---|
| SPA | GPP | 0.59 | 2.57 | -0.81 |
| | ET | 0.48 | 1.23 | -0.90 |
| ACM-GPP-ET | GPP | 0.61 | 2.41 | -0.66 |
| | ET | 0.58 | 1.07 | -0.77 |

## 5.2 FLUXNET2015: Independent validation

For phase 2) validation ACM-GPP-ET and SPA simulated GPP and ET estimates for 59 sites from the FLUXNET2015 database to provide fully independent validation of ACM-GPP-ET's ability to simulate real-world estimates but also of its predictive skill

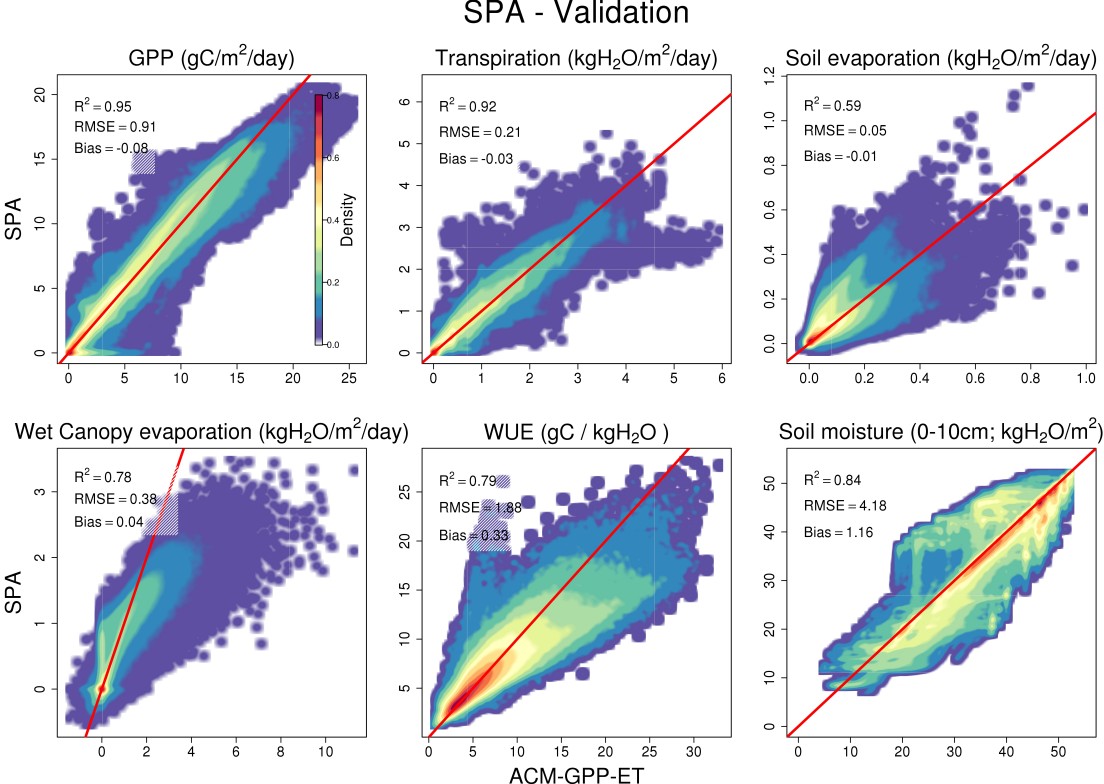

**Figure 4.** Comparison between ACM-GPP-ET and SPA simulated fluxes for model based validation of ACM-GPP-ET's ability to simulate coupled plant carbon and water cycle dynamics. ACM-GPP-ET uses the maximum likelihood parameter set from the calibration procedure. Fluxes compared are gross primary productivity ($gC/m^2/day$), transpiration ($kgH_2O/m^2/day$), soil evaporation ($kgH_2O/m^2/day$) and evaporation of water intercepted by the canopy ($kgH_2O/m^2/day$). Also shown is the soil moisture content in the 0-10 cm soil layer ($kgH_2O/m^2$) and water use efficiency (WUE = GPP / T; $gC/kgH_2O$). The red line is a 1:1 line for reference. The colour intensity from blue to red indicates the density of flux estimates within a given area, note that for clarify showing low density areas the density values are scaled by $density^{1/4}$.

compared to that of SPA (Table 4; Figure 5). For the FLUXNET validation we aim to achieve a similar degree of predictive capacity to existing remote sensing driven estimates. As stated earlier we hypothese that both ACM-GPP-ET and SPA will perform better at forest sites than in other ecosystems. GPP is typically better predicted than ET by both ACM-GPP-ET and SPA, which is expected given that ET is the combination of three evaporative fluxes (ET = $E_{trans}$ + $E_{wet}$ + $E_{soil}$). Both GPP and ET are underestimated by ACM-GPP-ET and SPA with larger RMSEs than found when comparing between ACM-GPP-ET and SPA simulations (Table 4). However, ACM-GPP-ET marginally out-performs SPA at most sites and for ET in particular. The between site distribution of $R^2$ and RMSE is skewed with a relatively small number of sites performing poorly (Figure 5).

For each metric shown ($R^2$, RMSE and bias) the distribution achieved by ACM-GPP-ET indicates a potentially greater degree of predictive skill at daily time step than SPA.

ACM-GPP-ET and SPA perform well at forested sites (except evergreen broad-leaf forests), with more variable performance at grassland, crop and savannah type ecosystems (Figure 6). However, both ACM-GPP-ET and SPA demonstrated a clear capability to simulate inter-site variation (i.e. the mean GPP and/or ET between sites) $R^2 \sim 0.94$ for both GPP and ET. Variation in predictive capability is not unexpected given that we use a single set of parameters for both models without site specific modifications.

ACM-GPP-ET simulated and Eddy covariance derived estimates of GPP and ET were compared at different temporal aggregations (weekly, monthly and annual) showing good skill at simulating seasonal and inter-annual dynamics. From daily through to weekly and monthly aggregation the statistical agreement between variation in simulated and observed estimated improves considerably. Estimation of ET is most improved increasing from $R^2 = 0.58 \rightarrow 0.72 \rightarrow 0.75$ while for GPP $R^2$ increases from $0.61 \rightarrow 0.65 \rightarrow 0.68$. RMSE and mean bias remain largely unchanged. However, simulation of inter-annual variation is more challenging with the $R^2$ for GPP = 0.68 and ET = 0.46. A similar pattern of results is found for SPA (not shown).

## 6   Discussion

In this study we have described, calibrated (Figure 3) and validated (Figure 4-7) a model of intermediate complexity, the Aggregated Canopy Model for Gross Primary Productivity and Evapo-Transpiration (ACM-GPP-ET v1). ACM-GPP-ET provides a process orientated representation of plant photosynthesis and water cycle, coupled through ecophysiological principles (Figure 1). ACM-GPP-ET simulations using a single global calibration have been validated against simulated GPP and evaporative fluxes but also emergent properties including WUE and soil moisture status from the state-of-the-art SPA model. These simulations were driven with LAI, fine root stock and meteorological conditions spanning across global gradients (Figure 2). Furthermore, to provide fully independent validation we have compared our estimated GPP and ET fluxes, again using a single global calibration, against multiple eddy-covariance-derived flux data from the FLUXNET2015 database, demonstrating substantial predictive skill.

### 6.1   ACM-GPP-ET: lessons learned on model simplification

A number of alternate model structures were tested over the course of the development of ACM-GPP-ET, and while it is out of scope to describe these in detail, there are a range of important lessons learned from development of specific components. The single most computationally expensive component is the iterative solution linking photosynthesis and transpiration via stomatal conductance. However, a coupled representation of stomatal conductance linking these processes was essential for maintaining predictive capacity of both canopy exchanges and emergent properties of WUE and soil moisture status. Similarly, the simulation of soil moisture dynamics is time-consuming, due to the need for simulating non-linear drainage processes occurring at sub-daily time steps. Originally a single layer bucket was tested but was unable to generate reasonable soil moisture dynamics and ultimately drought responses compared with SPA. Our experience is consistent with other studies which have explicitly

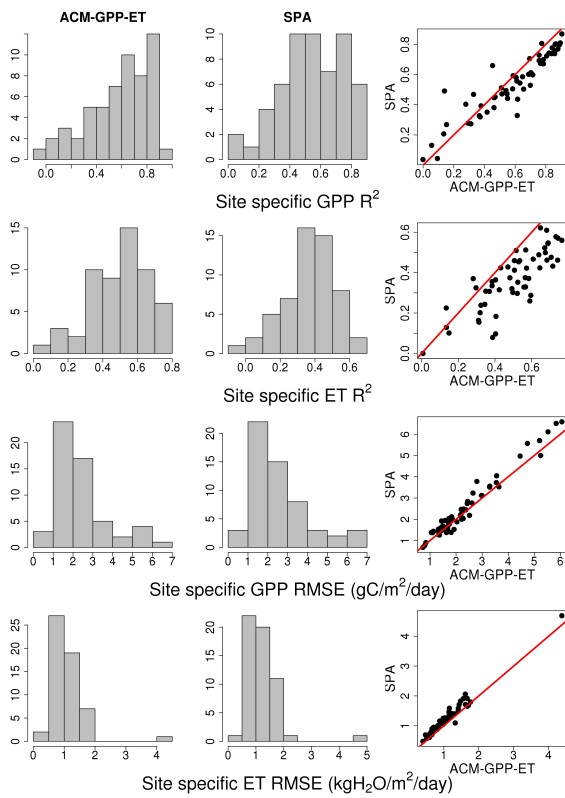

**Figure 5.** Comparison of ACM-GPP-ET and SPA simulating fully independent gross primary productivity (GPP) and evapotranspiration (ET) from FLUXNET2015. The histograms show the site specific $R^2$ and RMSE. SPA is shown in the left column, ACM-GPP-ET is in the centre. In the right column shows the site by site comparison of statistical metrics achieved by each model.

considered the impact of varying the number of soil moisture layers (Blyth and Daamen , 1997). Water drainage between soil layers and runoff of water from the canopy surface places an upper limit on efficiency achievable while maintaining predictive skill for soil moisture status and indirectly canopy fluxes. However, we expect further efficiency improvements to be achievable through subsequent code modifications including alternate theoretical approaches to achieve the photosynthesis-transpiration coupling. A dedicated focus on code optimality is out of scope for the current study, but is critical to the ongoing process of model improvement.

In this study 22 parameters are calibrated (Table 3), 15 of these are related to the estimation of canopy and / or soil absorption of PAR, NIR and longwave radiation. The key challenge for the radiative transfer was the essential requirement to reproduce the emergent non-linear functional shape between LAI and canopy radiation absorption, transmittance to soil and reflectance making the complex vertical structure implicit in the calibration. We found an appropriate simulation of non-linear radiative transfer was critical for realistic radiative responses of each component of evaporation. In contrast, for a GPP model alone

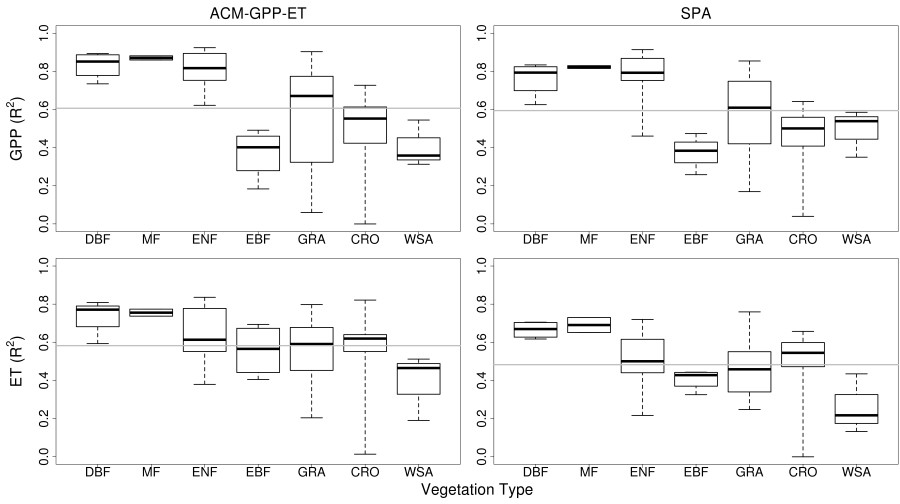

**Figure 6.** Independent validation of ACM-GPP-ET and SPA using eddy covariance derived estimates of gross primary productivity (GPP) and evapotranspiration (ET) from the FLUXNET2015. Box and whisker plots show the site specific $R^2$ values for each of the International Geosphere-Biosphere Programme (IGBP) vegetation classification. CRO = cropland, CSH = closed canopy shrubland, DBF = deciduous broad-leaf forest, EBF = evergreen broad-leaf forest, ENF = evergreen needle-leaf, GRA = grassland, MF = mixed forest, WSA = woody savanna. The mean for each vegetation classification is shown by the thick centre line, the box represents the 25 % and 75 % quantiles while the whiskers indicate the minimum and maximum values. The horizontal grey line indicates the mean $R^2$ across groups.

a far simpler radiative transfer scheme was viable(Williams et al., 1997). However, the large number of parameters in the radiative transfer scheme is open to constraint through e.g. remote sensing observations of canopy structure and reflectance. These observations could be used to calibrate the scheme for individual locations but also canopy structural forms (i.e. canopy vertical structure).

## 6.2   Intermediate complexity emulation of a complex TEM

ACM-GPP-ET accurately simulates its calibration data-set (Figure 3) and out-of-sample validation (Figure 4) generated by the SPA model. Substantial predictive skill was achieved for photosynthesis and each of the evaporative fluxes, but also the unbiased simulation of soil moisture and WUE which were not part of the calibration process (Figure 4). The single global calibration effectively spans global process sensitivity to climate across gradients of latitude, maritime-continental gradients, and seasonal cycles. The calibration also represents the effect of ecological variation of LAI and fine root stocks. By including 10 different drivers we generated a major challenge for model simplification. ACM-GPP-ET must robustly represent functional forms for C and water cycling across these multiple response dimensions, including any interactions. We note an underestimate in peak transpiration fluxes (Figure 3) which we hypothesise is due to the lack of including the impact of energy balance on canopy and non-linear responses at sub-daily timescales. While this bias may in some cases lead to an underestimate of within

day drought / water supply limitation the statistical analyses for validation indicate that the functional forms embedded in ACM-GPP-ET effectively represent those arising from complex mechanistic interactions within SPA. ACM-GPP-ET generates robust daily aggregations from SPA's hourly resolution.

ACM-GPP-ET effectively reproduces the eco-climatological sensitivity of plant water use efficiency from the SPA model (Figure 4). Reliable simulation of the dynamics and magnitude of plant WUE is an important property for robust modelling of hydrological, ecological and biological interactions. For climate sensitivity studies, ecosystem carbon-water coupling controls drought development and its interactions with ecosystem processes (Beer et al., 2009; Keenan et al., 2013; Bonan et al., 2014). Similarly, appropriate partitioning of evaporative fluxes, i.e. T/ET, is essential to simulate correctly the overall ecosystem response to change in climate. Transpiration has a direct interaction with biogeochemical cycling through canopy conductance, whereas evaporative fluxes have an indirect effect through adjustments to soil moisture and radiation environment mediated through variation in canopy cover. T/ET estimated by ACM-GPP-ET and SPA are closely aligned for both the calibration procedure with fixed soil moisture and that with dynamic soil moisture. The fraction of T/ET declined under dynamic soil moisture in both models, indicating consistent response to varied water status. This consistency indicates that ACM-GPP-ET effectively represents and aggregates to canopy scale the critical nexus for carbon and water cycling simulated in the leaf-scale stomatal conductance routines of SPA. As noted in section 6.1 alternate coupling approaches could be investigated reduce the computational requirements, but perhaps more importantly our framework allows for the testing of alternate coupling hypotheses to robustly assess their predictive capacity against emergent properties For example, recent analyses estimate transpiration is responsible for ∼58 % of global evaporation, consistent with both SPA and ACM-GPP-ET estimates, and contrasting with many TEMs which tend to underestimate T/ET partitioning (Wei et al., 2017) but also expanding on the stomatal model comparison conducted by Bonan et al. (2014).

### 6.3   Comparison to independent flux observations worldwide

A key component of our validation process was the comparison of ACM-GPP-ET and SPA simulated GPP and ET fluxes at 59 FLUXNET2015 sites. We aimed to achieve a comparable predictive skill compared to analyses driven by remotely sensed information. Moreover, we hypothesised that both ACM-GPP-ET and SPA would perform best at forest sites due to the choice of forest based parameters used in the calibration procedure. ACM-GPP-ET and SPA simulated both GPP and ET fluxes with substantial skill, especially given that only a single global parameterisation was used (Table 4; Figure 5-6). Indeed, ACM-GPP-ET slightly out-performs SPA in each statistical metric presented here; the greatest difference is found for simulating daily variation of GPP and in particular ET fluxes (Figure 5). However, it is unlikely that ACM-GPP-ET has actually improved on SPA itself, as ACM-GPP-ET is an emulation of SPA, therefore the difference should not be viewed as significant. The improved statistics found for ACM-GPP-ET are likely due to a combination of factors underlying errors which by chance lead to an apparent improvement. One exception to this assumption is that SPA's sub-daily meteorological drivers were gap filled based on down-scaled reanalysis drivers (as used in the calibration process) which as noted in the introduction can introduce errors comparable in magnitude to the direct daily aggregation (e.g., Williams et al., 2001a). Finally, as hypothesised SPA and ACM-GPP-ET performed best at forests sites, with some grassland, cropland and woody savanna sites performing less

well (Figure 6). The relative pattern of performance between vegetation types is consistent between ACM-GPP-ET and SPA, strongly indicating consistent underlying response to a wide range of climate conditions and ecological states, and similar predictive capabilities when applied in circumstances without site-specific information.

The ACM-GPP-ET predictions used a single, global calibration, and thus evaluated a single response surface to FLUXNET data, without taking into account any ecological variation in plant processes among FLUXNET sites beyond LAI, fine root biomass and soil textural information (potential errors in these components is discussed below). Thus, critical plant traits, such as the maximum rate of carboxylation per leaf area (Vcmax), stem hydraulic conductance ($G_p$) and rooting depth, were set the same across all sites for SPA and ACM-GPP-ET. But, we expect these parameters to vary, given our knowledge of trait variation at sites and from worldwide studies (Wright et al., 2004; Kattge et al., 2011; Johnson et al., 2018). So, the global parameterisation is likely to be biased for most sites. The poorer performance of SPA and ACM-GPP-ET in predicting GPP in evergreen deciduous forests is likely linked to a bias in parameters, for example (Figure 5). For a more robust global application, ACM-GPP-ET requires prior estimates for local values of the parameters in Table 3. Based on previous works by Bonan et al. (e.g., 2014) we expect the most important local parameterisation will be for root resistivity, plant conductivity and NUE. Root resistivity and plant conductance determine the maximum rates of water transport to the canopy. NUE determines the capacity of carboxylation and electron transport in photosynthesis.

Errors in the LAI, fine root biomass and soil textural information used in the SPA and ACM-GPP-ET model inputs are also likely. Errors in the remote sensing of LAI assimilated within the CARDAMOM framework will propagate into the resulting estimates of LAI and fine root biomass. Soil textural information comes from the HWSD database the errors for which are poorly described. Furthermore, we made assumptions about root depth and canopy height which are also likely biased. LAI information ultimately comes from MODIS (or other satellite products) which has varied skill depending on vegetation cover type (Yan et al., 2016). Yan et al. (2016) showed MODIS LAI detects LAI dynamics least well over forests (particularly needle-leaf), but perhaps more critically for our analysis they also showed a consistent RMSE between 0.6-0.8 m$^2$/m$^2$ between vegetation types. For arable crop land and grassland such an RMSE could constitute an error in the magnitude of LAI on the order of 66 % (in their observation dataset), potentially resulting in substantial errors and bias in estimation of ecosystem fluxes.

## 6.4 Comparison to other global GPP and ET estimates

We aimed for and achieved ACM-GPP-ET's performance against FLUXNET2015 sites to be comparable to that demonstrated by GPP and ET estimates generated by a range of alternate approaches and temporal resolutions (e.g., Jung et al., 2011; Mu et al., 2011; Martens et al., 2017; Wang et al., 2017). For example, FLUXCOM uses a machine learning approach assimilation and a wide range of global spanning information to estimate monthly GPP (Number of sites not reported, R$^2$ = $\sim$0.82, RMSE = $\sim$1.18 gC/m$^2$/day) and ET (Number of sites not reported, R$^2$ = $\sim$0.86, RMSE = $\sim$0.47 kgH$_2$O/m$^2$/day) (Jung et al., 2011). At monthly time scales FLUXCOM performs marginally better at estimation of variation in fluxes but with RMSEs roughly half that found with ACM-GPP-ET for both GPP and ET; this is not unexpected as FLUXCOM was calibrated against the FLUXNET database itself (Jung et al., 2011). Whereas the satellite-based remotely sensed derived 8-day MODIS estimates,

based on absorbed radiation and empirical response functions, perform less favourably than ACM-GPP-ET for both GPP (18 sites, $R^2 = 0.52$, RMSE = 0.96 $gC/m^2/day$; Wang et al., 2017) and ET (46 sites, $R^2 = 0.65$, RMSE = 0.84 $kgH_2O/m^2/day$; Mu et al., 2011). Finally, GLEAM estimates of ET performed similarly to those achieved by ACM-GPP-ET; GLEAM makes use of a comparatively complex approach to estimate ET, using a model of ecosystem water cycling updated by satellite based

remotely sensed information within a data assimilation framework to generate a daily estimate of the global water budget (63 sites, $R^2 = 0.64$, RMSE = 0.73 $kgH_2O/m^2/day$; Martens et al., 2017). In each case the approaches highlighted above made use of vegetation-type-specific information or location-specific remotely sensed biophysical information to drive their analysis compared to our comparatively naive approach using a single set of ecophysiological parameters. Therefore, we reasonably expect that significant improvements through the inclusion of location- and / or vegetation-type-specific calibration as would

be achieved through model-data fusion approaches (e.g.,  Bloom et al., 2016). A key benefit of our process-orientated approach is that unlike the above described approaches, which are each dependent on the input of remotely sensed information, our modelling framework can be used predictively to extrapolate into space and times where remotely sensed information are not available. This is particularly true when ACM-GPP-ET has been coupled to our DALEC C-cycle model allowing for feedbacks between carbon supply and available LAI. Further details on the potential of this coupling below.

## 6.5   Global Applications

It is typical to generate regional and global estimates of carbon and water cycling using complex land surface models. Such models make vital contributions to assessments of the global carbon budget (Le Quéré et al. , 2015) and weather and climate forecasts. A challenge for these models is that their complexity generates high computational demand, and they have demanding parameterisation needs. Thus, these models are often applied using plant functional type approaches, whereby parameters are

set for an entire biome, with no variation and no uncertainty is attached. There is a need then for models of intermediate complexity that are less demanding computationally, have fewer parameters, but retain realism.

ACM-GPP-ET is such a model, constructed from the simplification of a complex land surface model with a long evaluation history, SPA. ACM-GPP-ET captures the critical functional forms in carbon-water interactions the emerge from process representation at sub-canopy, hourly timescales. ACM-GPP-ET can represent the interactions of supply and demand on stomatal

opening, and how this responds to changes in atmospheric conditions and soil moisture states. This level of detail is critical for application in global change analyses that are vital for diagnosing and predicting earth system evolution. Thus, ACM-GPP-ET produces realistic outputs, based on comparison with its more complex pre-cursor.

ACM-GPP-ET is well suited for ensemble modelling schemes due to its faster run-time, as shown in the MH-MCMC calibration process used here with SPA outputs used as training data. The parameter posteriors generated here (Figure A1) provide

a starting point for full carbon cycle and water cycle analyses across regional to global domains. For instance, Bloom et al. (2016) have shown how an IC GPP model, ACM-GPP (Williams et al., 1997), combined with a carbon cycling model (DALEC; Williams et al., 2005), can be linked into a model-data fusion framework, CARDAMOM. CARDAMOM can, when combined with DALEC, retrieve probabilistic estimates of carbon stocks, fluxes and model parameters (including key unknowns such as photosynthate allocation to plant tissues and their residence times). CARDAMOM can produce outputs across a domain at the

resolution of input forcing (climate data, burned area) and observational constraints (satellite time series of LAI, biomass maps, soil C maps). The advantage of CARDAMOM is that it generates likelihoods for model initial conditions and parameter values that are consistent with climate forcing and domain observations from e.g. satellites, and their estimated errors. Currently CARDAMOM infers water limitations to C cycling through satellite observations of greenness alone. Because there is no coupling to a local water model, CARDAMOM cannot use modelled information on water balance, or independent observations such as surface soil moisture (e.g., Chen et al., 2018). Through using ACM-GPP-ET in CARDAMOM, it will be possible to assimilate new observational data related to water fluxes and state variables.

The combination of ACM-GPP-ET, coupled to DALEC, and CARDAMOM provide multiple direct and indirect avenues for propagating information acquired using intermediate complexity models to complex state-of-the-art TEMs. ACM-GPP-ET and SPA directly share 5 parameters calibrated in this study (Table 2) plus a further 9 biophysical traits which where not calibrated in this study (Table 3). Moreover, all of the parameters calibrated in this study (Table 2) can be indirectly related to those used in SPA (and many other TEMs) e.g., NUE which is closely related to Vcmax, Jmax and foliar nitrogen, but also radiation absorption/reflectance as a function of LAI. Similarly, when ACM-GPP-ET is combined with DALEC and used within the CARDAMOM framework analyses such as those carried out by Bloom et al. (2016) (as is intended) retrieving information on carbon stocks, carbon allocation and residence times results in the retrieval of ecologically relevant traits. These traits can be directly related to parameters found in most state-of-the-art TEMs equipped with a C-cycle. Such information should at a minimum provide information on spatial variation expected and in the optimum situation inform on the exact magnitude of those parameters.

### 6.6 Further opportunities and gaps

There are an array of next steps to undertake for further development both as a stand alone tool and as part of a coupled modelling framework along with DALEC and CARDAMOM. The ACM-GPP-ET parameters estimated here against SPA can be calibrated individually at FLUXNET2015 site (were sufficient biophysical information are available) to determine critical parameter variability to explain observed differences in fluxes. Driven with remotely sensed LAI ACM-GPP-ET could make global estimates of GPP, ET and WUE for direct comparison with outputs from FLUXCOM, GLEAM and CMIP5 model ensembles. As part of the CARDAMOM framework a site specific FLUXNET2015 analysis allows us to assess our ability to retrieve information on the whole carbon cycle across ecological and climate gradients within a data-rich environment, including key unknowns such as rooting depths which play a critical role in ecosystem resilience to drought. Such analyses provide the supporting frameworks needed to conduct global scale reanalyses and potentially near-term (next 12 months) and intermediate term (next 10 years) predictions with fully resolved uncertainties due to the propagation of ensembles.

Due to lack of space we have not reviewed the uncertainty estimates on parameter retrievals in the calibration of ACM-GPP-ET from SPA, but these contain useful information on the relative uncertainties in key processes in the aggregation. There are gaps in the capacity of ACM-GPP-ET globally; C4 pathways have not been included, nor organic soils. However, SPA is capable of simulating flux responses to these process adjustments, so new calibrations and/or model structure can be generated following a similar approach to that laid out here.

## 7 Conclusions

We have calibrated and robustly validated a model of intermediate complexity ACM-GPP-ET demonstrating good capacity of simulating the carbon-water dynamics of a state-of-the-art SPA model. ACM-GPP-ET has demonstrated substantial predictive skill when simulating fully independent eddy covariance derived estimates of carbon and water exchange which is comparable to that of other globally used GPP and ET products. Finally, ACM-GPP-ET is highly computationally efficient, $\sim$2200 times faster than SPA, opening up a substantial range of further opportunities.

*Code availability.* The code for ACM-GPP-ET version 1 has been made freely available from the Edinburgh DataShare (doi: http://hdl. handle.net/10283/3237). Subsequent source code developments will be made available under GNU General Public License (GPL) via Github (link will be made available in the final published manuscript). The Fortran source code for the Weather Generator v1 used when downscaling daily meteorology to hourly time step is available here https://github.com/GCEL/WeatherGenerator_v1.

## Appendix A: Appendix A: Calibrated parameter distributions

The CARDAMOM calibration process of ACM-GPP-ET retrieves multiple parameter sets consistent with the calibration dataset, resulting in a probability density function (PDFs) for each parameter. A detailed discussion of the PDFs retrieved is out of scope for this study, however a brief description of the primary features is given below (Figure A1).

The width (relative to the prior range) and overall shape of the PDF (i.e. uni- or multi-modal) gives an indication of the constraint achieved. The majority of parameter posteriors (16 out of 22) cover less than 50 % of their prior range and show a single clear peak value without substantial skew (e.g. NUE; Figure A1). Notable exception include aPAR_trans and kPAR_refl, indicating that there is a degree of equifinality in absorption the required PAR to support photosynthesis and canopy evaporation. Therefore, providing a potential focus for refinement of the model or calibration process (e.g. through the introduction of new data streams).

## Appendix B: Appendix B: SPA ecophysiological parameters

*Author contributions.* TLS coded the model and conducted the analysis. Both TLS and MW developed the experimental design and wrote the manuscript

*Competing interests.* The authors declare that they have no competing interests.

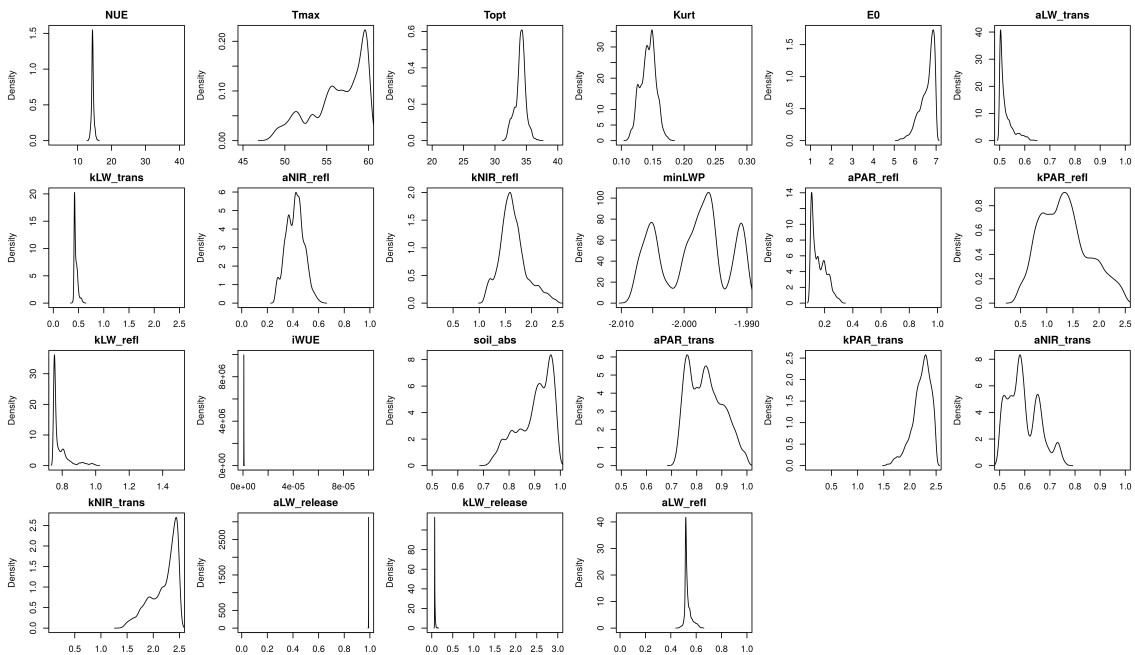

**Figure A1.** Probability density functions of the retrieved parameters for ACM-GPP-ET. The figure label refers to the name given in Table 1 of the main text. The range of the x-axis matches that of the parameter prior ranges to allow easy identification of those parameters which are the easiest to constrain.

**Table B1.** Ecophysiological parameters used by SPA within this study but not already provided in Table 2 and 3. These parameters are drawn from previous SPA publications (Williams et al., 1996; Bonan et al., 2014).

| Symbol | Value | Units | Description |
|--------|-------|-------|-------------|
| $\kappa C$ | 33.6 | $\mu$molC/gN/s | Coefficient relating the maximum rate of carboxylation to leaf nitrogen content |
| $\kappa J$ | 53.8 | $\mu$molC/gN/s | Coefficient relating the maximum rate of electron transport to leaf nitrogen content |
| $minLWP_{s}pa$ | 2 | MPa | Absolute value for minimum tolerated leaf water potential |
| $Leaf_{c}ap$ | 2500 | mmolH$_2$O/m$^2_{leafarea}$/MPa | Leaf / plant water capacitance for supply versus demand calculation |
| $leaf_{p}arrefl$ | 0.16 | fraction | Leaf level reflectance of incident photosynthetically active radiation |
| $leaf_{p}artrans$ | 0.16 | fraction | Leaf level transmittance of photosynthetically active radiation |
| $leaf_{n}irrefl$ | 0.43 | fraction | Leaf level reflectance of incident near-infrared radiation |
| $leaf_{n}irtrans$ | 0.26 | fraction | Leaf level transmittance of near-infrared radiation |
| $soil_{p}arrefl$ | 0.03 | fraction | Reflectance of photosynthetically active radiation incident on soil surface |
| $soil_{n}irrefl$ | 0.02 | fraction | Reflectance of near-infrared radiation incident on soil surface |

*Acknowledgements.* This work used eddy covariance data (FLUXNET2015) acquired and shared by the FLUXNET community, including these networks: AmeriFlux, AfriFlux, AsiaFlux, CarboAfrica, CarboEuropeIP, CarboItaly, CarboMont, ChinaFlux, Fluxnet-Canada, GreenGrass, ICOS, KoFlux, LBA, NECC, OzFlux-TERN, TCOS-Siberia, and USCCC. The ERA-Interim reanalysis data are provided by

ECMWF and processed by LSCE. The FLUXNET eddy covariance data processing and harmonization was carried out by the European Fluxes Database Cluster, AmeriFlux Management Project, and Fluxdata project of FLUXNET, with the support of CDIAC and ICOS Ecosystem Thematic Center, and the OzFlux, ChinaFlux and AsiaFlux offices. The study has been supported by the TRY initiative on plant traits (http://www.try-db.org). The TRY initiative and database is hosted, developed and maintained by J. Kattge and G. Bönisch (Max Planck Institute for Biogeochemistry, Jena, Germany). TRY is currently supported by DIVERSITAS/Future Earth and the German Centre for Integrative Biodiversity Research (iDiv) Halle-Jena-Leipzig. This work was funded primarily through the NERC GHG program GREENHOUSE project (grant NE/K002619/1) and the UK Space Agency Forests2020 project (https://ecometrica.com/space/forests2020). Additional resources were provided by the NERC National Centre for Earth Observations (NCEO).

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
