# Peer review of "Description and validation of an intermediate complexity model for ecosystem photosynthesis and evapo-transpiration: ACM-GPP-ETv1"

_Geoscientific Model Development, 2018_

## Referee Comment (RC1) · Martin De Kauwe (Referee) · 13 Feb 2019

Smallman and Williams describe a daily, canopy-scale, coupled photosynthesis-ET model, which they categorise as "intermediate" complexity. They ask how computationally efficient their simpler model is and whether this approach can adequately simulate the complex TEM. They also discuss potential research applications of this model.

The bottom line is that I think this paper is interesting, thorough and I can already

see many future applications - it warrants publication. However, I think the current presentation sells it a little short.

For example, the paper starts out by make strong claims for the need for a computational efficient model - fine. But if that is the desire, why not simply use a big-leaf/2-leaf, coupled A-ET model? Or even more simply, a linear approximation (e.g. Best et al. 2015J. Hydrometeor., 16, 1425–1442), or something between those two approaches involving machine learning? My guess, is that the reason is that speed isn't the ultimate objective. So being told that the new model is ∼2200 times faster than SPA, to me, isn't all that interesting. I bet I could make a simpler model than this which is quicker still. I'm not arguing that the choices made by Smallman and Williams aren't perfectly legitimate (although "fewer parameters than typical leaf-scale stomatal models" - is that actually true?); however, they aren't simply with speed in mind.

I would like to see more text devoting to the justification of their intermediate model. What is the underlying performance expectation here ("ACM-GPP-ET and SPA simulated GPP and ET fluxes at 59 FLUXNET2015 sites with substantial skill")? And how is the reduction in complexity affecting performance? My sense as a reader is I don't really know the answer to this question after reading the results. In Fig 3, the simpler model produces an R2 > 0.81 for all four fluxes. How much more would this have been degraded by simplifying the model further? Or phrased another way, which of the key assumptions are responsible for this performance? Knowing that might be really insightful for model development and a broader audience than users of the ACM/SPA models. It might also help identify areas where the emulator model could be improved further to be more mechanistic.

Finally, the authors clearly see model-data fusion as a means to calibrate such a model. Therefore, I wonder about some of the choices in terms of mechanism (see point about plant hydraulics).

Introduction ————

[Figure]

- Pg 2, line 23: This statement about TEMs being expensive (slow) requires some quantification. It is not my experience that standard TEMs, which simply solve coupled C-water fluxes are actually all that slow. No doubt a model such as SPA (that the second author works with) is undoubtedly slower than most TEMs, but as a blanket statement?

- Pg 2, line 24: the text around issues to do with model data fusion ignores recent advocates of emulators (e.g. Fer, et at. Biogeosciences, 15, 5801-5830, 2018.).

- Pg 2, line 27-30. Whilst this is a valid argument, I wonder what the evidence is that use of a daily model is less biased than a sub-daily model reliant on a weather generator? This text also ignores a number of papers that have attempted to approximate sub-daily behaviour without the need for a weather generator (e.g. Sands, P. J. (1995). Australian Journal of Plant Physiology 22, 603-6 14.).

- Pg 2, Line 31: Agreed and I note that the second author has already produced such a model, ACM. I suggest some text at this point to discuss this and how the proposed approach differs is warranted. Presumably, the distinction is the coupling of the carbon and water cycle and I suggest it is worth including the history of ACM in the discussion on Pg 3.

Methods: ———

- What is the link between eqn 1, Pn and gc, eqn 2? Surely Pn should depend on gc? In fact, the final eqn for GPP, number 14, which is dependent on gc makes sense, but what is the connection to eqn 1 and where is this explained?

- Where are the equations for gs and gb? I actually see these are included below eqn 57 for gb. It would be worth telling the reader this at the point gs and gb are introduced.

- Page 8, line 5: why is the reference temp 20 and not 25 degrees?

- How is the isothermal net radiation estimated?

- Why add in the complexity of the optimisation (sec 2.6) and/or the plant hydraulic resistance (2.9)? Surely this isn't more efficient than a simpler bucket type soil model? Further, in 2.9, the plant hydraulics resistance actually makes little use of commonly measured traits (e.g. p50). I wonder if this isn't quite a disadvantage moving forward. What would any proposed optimisation scheme be calibrating against?

Results: ——-

- I feel like the start of the results could benefit from a sentence introducing what is happening again. At this point, the manuscript is 22 pages long and although the CAR-DAMON stuff was introduced in the methods, I suspect you could forgive the reader for being a bit lost. My suggestion would be to re-read 5.1 standalone and see how clear it is for a reader, I would suggest it could be revised.

- Pg 22: Do the authors have thoughts on why the model is underestimating peaks in transpiration as simulated by SPA?

- Pg 23: "However, ACM-GPP-ET marginally out-performs SPA at most sites and for ET in particular." - how should this be interpreted by the reader? My interpretation is that the simpler model, which is a calibration ought not to out perform SPA and if it does so, it does so for the wrong reason. This warrants some comment.

- I didn't find the comparison in Fig 5 particularly insightful. Lumping all the sites means that we don't learn anything. Where does the model perform best, worst? What does this performance tell us about the underlying mechanisms? In that sense, Figure 7 is more useful and perhaps 5 could be omitted?

---

## Referee Comment (RC2) · Anonymous Referee #2 · 20 Feb 2019

The Smallman et al Paper describe a new intermediate complexity model of photosynthesis and evapotranspiration. Such model is partly processes based but using a single canopy layer and daily output is 2000 time faster than full process based models based on half-hourly time steps. The paper is well written. The description of the model, the calibration and validation procedure are sound. The model show good performances compared to the SPA model used for calibration. So I have only few remarks on the model description. However what is missing here is what is really the added value of such a model and what are the final objectives for the development of this model ? It is

stated in the abstract "that model of high complexity cannot be evaluated for their parameter sensitivity nor calibrated thru assimilation of large ensemble" but it is not clear how a model like ACM-GPP-ET can solve this problem ? Indeed, they are not based on the same parameters and set of equations so how to infer sensitivity of parameters from the complex model from sensitivity of parameters from intermediate complexity model ? Likewise how calibration of intermediate complexity model helps to calibrate complex model (especially in the case on the paper it is the opposite). Likewise authors compare results for ACM-GPP-ET to different learning machine algorithms showing similar skill. But then what is the added values of such model compare to machine learning approaches ? For me machine learning algorithms, are very well suited to interpolate informations but are difficult to use outside of their domain of training set. On the opposite model, based on process can be extrapolated. Then such model could be used for past or future climate conditions for instance. But ACM-GPP-ET is not a full vegetation model. In particular, it depend of LAI that should be prescribed and not calculated. So simulation are limited to conditions where LAI observations are available. So it would be interesting to know if the final objective is to include such kind of model in a TEM to be able to simulate the complete carbon cycle. In summary the description of the model and its validation is correct. The model give very satisfying result. But what is really missing in the paper is what are the objective and perspective for such a model.

For more specific points:

- One argument for the intermediate complexity model is the difficulty to have sub-daily climate forcing. I am not very convinced since, first of all most of global products are available at 3 to 6 hourly time steps. Then, even for data only available at daily time-steps, most of the TEM use weather generators to simulate a pseudo diurnal cycle. So this is not really a limit. An interesting question could be to know if intermediate complexity model directly based on daily data are performing better with daily data that more complex models coupled with a weather generator ? The paper partly reply to this point in the comparison between SPA and ACM-GPP-ET on fluxnet site where ACM-

GPP-ET seems to perform a little better that SPA. However the difference is relatively small. This first point brings me to a second one about FLUXNET. As ACM-GPP-ET is calibrated on SPA, it will obviously tend to have the same behaviour and then the same discrepancies when compare to FLUXNET (which is visible on Figure 7).So as suggested in the further opportunities it would be interesting to make a calibration of ACM-GPP-ET on observed GPP and ET from fluxnet to see how the calibration differ and how model improve compared to fluxnet.

- I am a little surprised by the calibration protocol making a simulation from 2001 to 2012 with rapid increase of CO2 ? Since I guess there is some prognostic variables it means that there is correlations between the successive years of simulation and then the rapid (and unrealistic) change in CO2 could lead to artefact in the simulation. So to explore the range of CO2 why not doing a series of simulations from 2001 to 2012 with different (but fixes) levels of CO2 ?

- The way LAI is used in the calibration is not clear. It is stated that it is retrieved from DALEC and few lines after it is stated that CARDAMOM assimilate MODIS LAI ? So which LAI is used everyday to force ACm6GPP-ET ? And is the LAI from DALEC is retrieved from a previous simulation or done with a simulation with the same Era-iterim forcing (and CO2 increase) ? this point must be clarified

- in equation 12 use P to define day length where different Pn,Pd,Pi represent different GPP limited term is not very appropriate !

---

## Author Response (AR1)

**Author's response to review of "Description and validation of an intermediate complexity model for ecosystem photosynthesis and evapo-transpiration: ACM-GPP-ETv1"**

Reviewer comments are written in normal black text, authors responses are written in blue while new text in the manuscript in *italics*.

We thank both reviewers for their thorough and encouraging comments on our manuscript. We broadly agree with the comments given and have worked to revise the manuscript to bring clarity to our objectives, explain the novelty of our approach and to highlight the pathways of information transfer between our modelling tools and useful ecological information.

In terms of our motivation, we note that models are often updated with added complexity, whereas we should challenge ourselves to create simpler models. Simpler models are: 1) faster running (fewer calculations); 2) easier to understand (less code); 3) more focused (fewer routines). Simplification means models can be more thoroughly explored, shared, calibrated and tested. Further, we are interested in models with ecological relevant parameters and state variables, such as rooting depth, hydraulic resistance, and LAI. Parameters that are measurable from space, or have values in plant trait databases that can serve as priors, are particularly valuable and were prioritised in our activity here.

**Reviewer 1: Martin De Kauwe**

Smallman and Williams describe a daily, canopy-scale, coupled photosynthesis-ET model, which they categorise as "intermediate" complexity. They ask how computationally efficient their simpler model is and whether this approach can adequately simulate the complex TEM. They also discuss potential research applications of this model.

The bottom line is that I think this paper is interesting, thorough and I can already see many future applications - it warrants publication. However, I think the current presentation sells it a little short.

For example, the paper starts out by make strong claims for the need for a computational efficient model - fine. But if that is the desire, why not simply use a big-leaf/2-leaf, coupled A-ET model? Or even more simply, a linear approximation (e.g. Best et al. 2015J. Hydrometeor., 16, 1425–1442), or something between those two approaches involving machine learning? My guess, is that the reason is that speed isn't the ultimate objective. So being told that the new model is ~2200 times faster than SPA, to me, isn't all that interesting. I bet I could make a simpler model than this which is quicker still. I'm not arguing that the choices made by Smallman and Williams aren't perfectly legitimate (although "fewer parameters than typical leaf-scale stomatal models" - is that actually true?); however, they aren't simply with speed in mind.

Thank you for your comment Martin. Indeed speed is not the only criterion we aimed to achieve with the development of ACM-GPP-ET. It is clear than we have not sufficiently brought forward our parallel objectives of a more computationally efficient modelling approach, but one that is

still process-orientated and capable of generating realistic emergent properties of the plant photosynthesis and water cycles.

We have modified the manuscript in multiple ways to bring forward both clarity on our objectives and the context for why we chose the approach we have used. The modifications include but not limited to:

Clarifying that there are multiple coupling points between the plant carbon and water cycles at both the stomata scale and also at the root-soil interface:

P2 L8-12: "Access to CO2 is controlled via leaf stomata, which provide the primary coupling point between GPP and the water cycle. Stomatal opening results in water loss via transpiration creating a dependency on accessible soil moisture, which is controlled by root biomass and its distribution through the soil profile. Thus, the root-soil interface is a second coupling point between the plant carbon and water cycles (Beer et al., 2009; Bonan and Doney, 2018)."

Clarify that complex models being slow is from the perspective of carrying out large ensemble based analyses:

P2 L24-26: "However, the increasing complexity of TEMs presents new challenges. Many of the most complex TEMs are too slow for use in model-data fusion analyses which are reliant on massive ensemble simulations (e.g., Ziehn et al., 2012; Smallman et al., 2017)."

Clarifying that any simplifications must still allow for realistic simulation of the interactions between plant carbon and water cycles to ensure appropriate simulation of emergent responses:

P3 L6-7: "The challenge here is to produce a model both sufficiently mechanistic to represent the coupling between plant carbon and water cycles linking to ecophysiological processes and observations of key global unknowns (e.g. rooting depth), but also computationally fast enough to be integrated into model-data fusion schemes and to allow a full exploration of parameter-related uncertainties."

Further we have variously modified paragraph three of the introduction to highlight some of the challenges surrounding even simpler approaches, such as over sensitivity to drought when using a single layer "bucket" and the breakdown in realism when comparing separate independently calibrated GPP and ET models. Thus, we emphasise the need to have the right estimates of GPP and ET for the *right* reasons.

P3 L8-24: "Photosynthesis is often estimated using physiologically realistic light, CO2 and temperature response functions (e.g., Jones, 1992; Williams et al., 1997). Evaporation is frequently estimated using simplified versions of the Penman-Monteith model, typically modelling plant stomatal regulation as a function of environmental drivers (e.g., Priestley and Taylor, 1972; Fisher et al., 2008). The impact of moisture limitations on both GPP and ET is

commonly achieved through the use of VPD as a proxy (e.g., Mu et al., 2011; Wang et al., 2017) or using a single soil layer "bucket" (e.g., Martens et al., 2017). While simple models can show skill when compared to in-situ estimates (Mu et al., 2011; Bloom and Williams , 2015; Martens et al., 2017; Wang et al., 2017) they usually estimate a single process, either photosynthesis or evapo-transpiration, neglecting their coupling. Without coupling, the feedbacks between C and water cycles will not be modelled robustly. For instance, there is a high risk that independently calibrated, simple GPP and ET models that are coupled naively in a plant-soil model framework will misdiagnose the sensitivity of water use efficiency (C fixed per water transpired) and have low predictive capability outside of the calibrated range (e.g., big leaf vs multiple leaf canopy; Tuzet et al., 2003; Wang and Leuning, 1998). Thus, connecting a series of simple models to generate a model of intermediate complexity (IC) carries significant risks. The IC model must represent process interactions effectively. A key test therefore is that any IC model must reproduce the sensitivities of key processes (i.e. GPP, ET,), their interactions (WUE) and soil moisture status demonstrated by the state-of-the-art TEMs, to ensure flux estimates are not only right but right for the right reasons"

**I would like to see more text devoting to the justification of their intermediate model.**

Furthermore, we have added a new paragraph specifically detailing the previous works upon which our current study was based. We include reference to gaps in our current modelling tools, linking to globally important unknowns, which ACM-GPP-ET is specifically intended to address:

P3 L25-P4 L8: "This study builds on two previously developed aggregated canopy models (ACM) for GPP (Williams et al., 1997) and ET (Fisher et al., 2008), and an existing state-of-the-art TEM SPA (Williams et al., 1996; Smallman et al., 2013). ACM-GPP simulated daily GPP sensitive to canopy nitrogen (N), temperature, absorbed shortwave radiation and atmospheric CO2 concentration; based on physiologically realistic relationships but lacking a representation of the impact of soil moisture availability on photosynthesis. Despite this limitation ACM-GPP has been coupled to the DALEC C-cycle model (Williams et al., 2005) and successfully used in model-data fusion experiments to improve our understanding of ecosystem C status, C allocation and residence times (Fox et al., 2009; Bloom and Williams, 2015; Bloom et al., 2016; Smallman et al., 2017) but also carbon-nitrogen interactions (Thomas and Williams et al., 2014). In addition to lacking a soil moisture response on photosynthesis, ACM-GPP limits the capacity of DALEC analyses to constrain the root component of the C cycle as roots currently play no ecological role within the modelling system (i.e. water or nutrient uptake). ACM-ET simulates the bulk ecosystem evapotranspiration based on a modified Penman-Monteith approach sensitive to absorbed shortwave radiation, temperature, vapour pressure deficit and wind speed. However, ACM-ET's bulk approach does not allow for distinguishing between different evaporative sources (i.e. soil surface, root extracted and canopy intercepted rainfall). Thus, it does not account for the different biotic and abiotic drivers which have varied responses to environmental change (Wei et al., 2017). Moreover, ACM-ET does not have a mechanistic coupling to water supply governed by root biomass and root vertical distribution. ACM-GPP and ACM-ET use different empirical models linking LAI,

minimum tolerated leaf water potential and meteorological drivers to estimate canopy conductance. Both ACM's can be calibrated to provide useful GPP and ET estimates, however when combined their predictive capacity for emergent properties such as WUE is limited (R2 < 0.2; data not shown) highlighting the need for further development to reproduce emergent ecosystem properties."

And how is the reduction in complexity affecting performance? My sense as a reader is I don't really know the answer to this question after reading the results. In Fig 3, the simpler model produces an R2 > 0.81 for all four fluxes. How much more would this have been degraded by simplifying the model further? Or phrased another way, which of the key assumptions are responsible for this performance? Knowing that might be really insightful for model development and a broader audience than users of the ACM/SPA models. It might also help identify areas where the emulator model could be improved further to be more mechanistic.

Thank you Martin for highlighting the need for clarity over what degree of predictive capacity is expected from the current model and what are the key development components of ACM-GPP-ET. We have addressed this in two components, first a clear statement of the expectations of the ACM-GPP-ET capacity in terms of validation against SPA and against the independent FLUXNET2015 information, and second what the lessons learned from the process.

Question 2 from the end of the introduction has been modified to remind the reader of the importance of key emergent properties not just GPP and ET fluxes.

P4 L30-31: "How well can the intermediate complexity ACM-GPP-ET emulate the complex TEM (i.e. GPP, ET, their coupling via WUE, and soil moisture)?"

We have modified the description of the validation process, Section 4, to improve clarity of the different phases of the validation process, their purpose and an explicit hypothesis regarding the performance against FLUXNET2015 sites. For example:

P23 L1-2: "Therefore, we hypothesise that both SPA and ACM-GPP-ET will perform best at forest sites and less well at sites with different hydraulic traits."

To highlight key learnings in the model development process Section 6.1 has been reformed to focus on "lessons learnt" rather than computational efficiency:

P28 L4-P29 L15: "A number of alternate model structures were tested over the course of the development of ACM-GPP-ET, and while it is out of scope to describe these in detail, there are a range of important lessons learned from development of specific components. The single most computationally expensive component is the iterative solution linking photosynthesis and transpiration via stomatal conductance. However, a coupled representation of stomatal conductance linking these processes was essential for maintaining predictive capacity of both

canopy exchanges and emergent properties of WUE and soil moisture status. Similarly, the simulation of soil moisture dynamics is time-consuming, due to the need for simulating non-linear drainage processes occurring at sub-daily time steps. Water drainage between soil layers and runoff of water from the canopy surface places an upper limit on efficiency achievable while maintaining predictive skill for soil moisture status and indirectly canopy fluxes. However, we expect further efficiency improvements to be achievable through subsequent code modifications including alternate theoretical approaches to achieve the photosynthesis-transpiration coupling. A dedicated focus on code optimality is out of scope for the current study, but is critical to the ongoing process of model improvement.

In this study 22 parameters are calibrated (Table 3), 15 of these are related to the estimation of canopy and / or soil absorption of PAR, NIR and longwave radiation. The key challenge for the radiative transfer was the essential requirement to reproduce the emergent non-linear functional shape between LAI and canopy radiation absorption, transmittance to soil and reflectance making the complex vertical structure implicit in the calibration. We found that an appropriate simulation of non-linear radiative transfer was critical for realistic radiative responses of each component of evaporation. In contrast, for a GPP model alone a far simpler radiative transfer scheme is open to constraint through e.g. remote sensing observations of canopy structure and reflectance. These observations could be used to calibrate the scheme for individual locations but also canopy structural forms (i.e. canopy vertical structure)."

Finally, the authors clearly see model-data fusion as a means to calibrate such a model. Therefore, I wonder about some of the choices in terms of mechanism (see point about plant hydraulics).

Our mechanistic focus was strongly prescribed by the structure of the complex TEM, SPA, that was the basis of the process modelling. SPA uses hydraulic functions to set potential and viable rates of liquid water flow, and links these to vapour phase losses. Hydraulic functions are now increasingly studied and linked to vegetation activity and climate sensitivity. Thus there is a developing dataset on hydraulics that the model can connect to.

**See responses to specific comments below.**

**Introduction ———**

- Pg 2, line 23: This statement about TEMs being expensive (slow) requires some quantification. It is not my experience that standard TEMs, which simply solve coupled C-water fluxes are actually all that slow. No doubt a model such as SPA (that the second author works with) is undoubtedly slower than most TEMs, but as a blanket statement?

Please see modified text indicated in the response to general comments above to statements on our intended meaning of "slow"

- Pg 2, line 24: the text around issues to do with model data fusion ignores recent advocates of emulators (e.g. Fer, et at. Biogeosciences, 15, 5801-5830, 2018.). (https://www.biogeosciences.net/15/5801/2018/bg-15-5801-2018.pdf)

You are quite right that alternate approaches are available and are entirely viable options. We have modified the manuscript to highlight some of the challenges surrounding alternate calibration approaches to our process emulation strategy:

P2 L26-29: "While effective and more computationally efficient alternative model-data fusion approaches are available they often reply on model code modifications, such as the creation of the model adjoint in variational approaches (e.g., Kuppel et al., 2012; Raoult et al., 2016), or model emulation often resulting in larger uncertainties in their posterior analysis (e.g., Fer et al., 2018)."

- Pg 2, line 27-30. Whilst this is a valid argument, I wonder what the evidence is that use of a daily model is less biased than a sub-daily model reliant on a weather generator?

Thank you for the comment we should have provided supporting evidence for this statement. The manuscript has now been modified to contain the following supporting evidence.

P2 L32-P3 L6: "Finally, there are major challenges in procuring sub-daily meteorological observations needed to drive TEMs away from meteorological stations - this is a particularly acute problem in tropical regions. Thus, TEMs are generally run using statistical down-scaled climate reanalysis data, which contain errors. The uncertainty generated when these errors are propagated into TEM GPP and ET estimates is comparable to IC model error associated with simulating daily fluxes directly(Williams et al., 1997, 2001a). Thus IC models have been shown to have similar errors to TEM models but at a lower computational cost (i.e. 1 time step verses 24 time steps) Thus, there is considerable value in having less complex, fast-running models that simulate GPP and ET."

This text also ignores a number of papers that have attempted to approximate sub- daily behaviour without the need for a weather generator (e.g. Sands, P. J. (1995). Australian Journal of Plant Physiology 22, 603-6 14.).

Unfortunately we are unable to gain access to the specific example paper. However, we are not aiming to generate sub-daily estimates so the concept of downscaling daily to sub-daily could be considered out-of-scope and potentially confusing to some readers of the current study. Moreover, the abstract for Sands (1995) makes clear that downscaling daily fluxes to sub-daily makes similar assumptions to downscaling meteorology to sub-daily. As discussed in the previous response, such downscaling introduces unavoidable uncertainty.

- Pg 2, Line 31: Agreed and I note that the second author has already produced such a model, ACM. I suggest some text at this point to discuss this and how the proposed approach differs is

warranted. Presumably, the distinction is the coupling of the carbon and water cycle and I suggest it is worth including the history of ACM in the discussion on Pg 3.

We agree with this suggestion. Please see the newly added paragraph 4 in the introduction for a detailed introduction of the precursor models.

**Methods: ------**

- What is the link between eqn 1, Pn and gc, eqn 2? Surely Pn should depend on gc? In fact, the final eqn for GPP, number 14, which is dependent on gc makes sense, but what is the connection to eqn 1 and where is this explained?

Pn refers to the metabolically limited photosynthesis, in the absence of CO2 limitations. This potential sink strength influences the CO2 gradient for exchange between the internal and external environments. This approach follows that used in the original ACM-GPP (Williams et al., 1997) and discussed in Jones (1992). The manuscript is updated in the following ways to make the clear.

**Opening to section 2.4:**

*"Following Williams et al. (1997) and Jones (1992), GPP is estimated as a co-limited function of temperature, CO2 (limited by stomatal opening and thus plant water availability) and absorbed PAR. "*

To improve clarity in the equations as indicated by both Martin and reviewer #2 the notation have been changed in the following ways (original -> revised)

 $P_n \rightarrow P_{NT}$  $P_d \rightarrow P_{co2}$  $P \rightarrow dayl$

Minor corrections to the notation have been made elsewhere in the manuscript and appropriately indicated in the tracked-changes version of the manuscript.

- Where are the equations for gs and gb? I actually see these are included below eqn 57 for gb. It would be worth telling the reader this at the point gs and gb are introduced.

The manuscript has been modified to refer the reader at this point to equation 57 for gb and section 2.6 for gs.

**- Page 8, line 5: why is the reference temp 20 and not 25 degrees?**

The reference value is as calculated in the source paper (McMurtie et al. (1992)) as stated in the manuscript.

P8L11-12: " $C_{comp}$  determines the  $C_i$  at which GPP becomes positive while  $C_{half}$  is the Ci at which  $CO_2$  limited photosynthesis is at 50% of its maximum rate. Both Ccomp and C half are calculated as a function of temperature following McMurtie et al. (1992)."

- How is the isothermal net radiation estimated?

Isothermal longwave radiation balance calculation is described in sec 2.7.2. The sub-heading as been modified to clarify this. We have also pointed the reader to this in P10 L14.

**- Why add in the complexity of the optimisation (sec 2.6) and/or the plant hydraulic resistance (2.9)? Surely this isn't more efficient than a simpler bucket type soil model?**

There are several reasons why we chose to use the iWUE optimisation approach to solve our stomatal conductance and to couple this model to multi-layered soil with water supply based on plant hydraulics. First, the iWUE approach, which is also used in SPA, was demonstrated to be more effective under drought conditions than Ball-Berry style approaches (Bonan et al., 2014; see section 2.6). This approach explicitly requires an estimate of water supply. One of the objectives, or rather desired outcomes of the development of ACM-GPP-ET is to connect the roots to photosynthesis to help constrain this component of the carbon cycle, as well as ecosystem traits such as rooting depth (see newly added paragraph 4 of the introduction) but also to link to novel data sources such as SapFIUXNET (P18L3-5). Finally, the choice to use multiple soil layers was driven by the development process. Originally a single layer bucket was tested but was unable to generate reasonable soil moisture dynamics and ultimately drought responses compared with SPA. Incrementally, additional layers were added to improve the soil moisture dynamics. The 4 soil layers is also supported by Blyth and Daamen (1997) who tested different number of soil layers for different soil textures with 4 layers being indicated as the best trade-off on model simplicity and effectiveness.

P28L13-P29L1: "Originally a single layer bucket was tested but was unable to generate reasonable soil moisture dynamics and ultimately drought responses compared with SPA. Our experience is consistent with other studies which have explicitly considered the impact of varying the number of soil moisture layers (Blyth and Daamen , 1997)."

Further, in 2.9, the plant hydraulics resistance actually makes little use of commonly measured traits (e.g. p50). I wonder if this isn't quite a disadvantage moving forward. What would any proposed optimisation scheme be calibrating against?

ACM-GPP-ET model structure relies on using hydraulic resistances, which include plant traits which exist within trait databases and are observable (e.g. root resistivity, stem conductance; see Table 3), to estimate the potential hydraulic flow related to the soil water potential and coupling ultimately to atmospheric demand to evaporation. While in this study the parameters listed in Table 3 are not calibrated by the MCMC analysis, they could in the future. Moreover, due to the mechanistic nature the model could estimate equivalents of other frequently measured information such as p50 as emergent properties. Through this avenue such information could be used to improve the calibration.

We believe that this concern has been addressed by the additional introduction provided by paragraph 4 in the introduction but also P18L3-5:

"The advantage to using a mechanistic approach allows for the estimation of physiological properties which makes possible novel comparisons with field observations such as Poyatos et al. (2016)."

**Results: -----**

- I feel like the start of the results could benefit from a sentence introducing what is happening again. At this point, the manuscript is 22 pages long and although the CARDAMON stuff was introduced in the methods, I suspect you could forgive the reader for being a bit lost. My suggestion would be to re-read 5.1 standalone and see how clear it is for a reader, I would suggest it could be revised.

We now begin the Results section with some summary test:

"We show that a single global calibration of ACM-GPP-ET can effectively reproduce the patterns of GPP and ET simulated by SPA. Importantly the predictions of WUE are consistent for both ACM and SPA, so that the simplified model is able to capture the interactions between C and water cycling. We also describe an independent validation against FLUXNET data, across 59 sites."

**- Pg 22: Do the authors have thoughts on why the model is underestimating peaks in transpiration as simulated by SPA?**

Implicit to their nature aggregate models are likely to underestimate extremes. In this instance we have the explicit hypothesis that these peaks are missed to due to the lack of energy balance closure of strongly non-linear responses which occur at sub-daily time-scales. We have added text to the discussion to reflect this.

Sec. 6.2: "ACM-GPP-ET must robustly represent functional forms for C and water cycling across these multiple response dimensions, including any interactions. We note an underestimate in peak transpiration fluxes (Figure 3) which we hypothesise is due to the lack of including the impact of energy balance on canopy and non-linear responses at sub-daily timescales. While this bias may in some cases lead to an underestimate of within day drought / water supply limitation the statistical analyses for validation indicate that the functional forms embedded in ACM-GPP-ET effectively represent those arising from complex mechanistic interactions within SPA. ACM-GPP-ET generates robust daily aggregations from SPA's hourly resolution"

- Pg 23: "However, ACM-GPP-ET marginally out-performs SPA at most sites and for ET in particular." - how should this be interpreted by the reader? My interpretation is that the simpler model, which is a calibration ought not to out-perform SPA and if it does so, it does so for the wrong reason. This warrants some comment.

We agree that this result does not indicate that ACM-GPP-ET is actually better than SPA, but rather is a result of both random error and potentially as a result of errors in gap-filling the

sub-daily meteorological drivers used in SPA. The manuscript has been modified to include the following in Section 6.3:

"Indeed, ACM-GPP-ET slightly out-performs SPA in each statistical metric presented here; the greatest difference is found for simulating daily variation of GPP and in particular ET fluxes (Figure 5-6). However, it is unlikely that ACM-GPP-ET has actually improved on SPA itself, as ACM-GPP-ET is an emulation of SPA, therefore the difference should not be viewed as significant. The improved statistics found for ACM-GPP-ET are likely due to a combination of factors underlying errors which by chance lead to an apparent improvement. One exception to this assumption is that SPA's sub-daily meteorological drivers were gap filled based on down-scaled reanalysis drivers (as used in the calibration process) which as noted in the introduction can introduce errors comparable in magnitude to the direct daily aggregation (e.g., Williams et al., 2001a)"

- I didn't find the comparison in Fig 5 particularly insightful. Lumping all the sites means that we don't learn anything. Where does the model perform best, worst? What does this performance tell us about the underlying mechanisms? In that sense, Figure 7 is more useful and perhaps 5 could be omitted?

We appreciate that this figure is not the most informative, figure 5 along with the density plot component was intended to provide continuity with the results presented in Figure 3 and 4. However, we agree that removing Figure 5 will allow for a smaller paper with a clearer message. The statistics presented alongside of Figure 5 will now be provided in a new Table 4. Figures 6 and 7 have correspondingly been re-numbered

**Anonymous Referee #2**

The Smallman et al Paper describe a new intermediate complexity model of photosynthesis and evapotranspiration. Such model is partly processes based but using a single canopy layer and daily output is 2000 time faster than full process based models based on half-hourly time steps. The paper is well written. The description of the model, the calibration and validation procedure are sound. The model show good performances compared to the SPA model used for calibration. So I have only few remarks on the model description. However what is missing here is what is really the added value of such a model and what are the final objectives for the development of this model ?

It is stated in the abstract "that model of high complexity cannot be evaluated for their parameter sensitivity nor calibrated thru assimilation of large ensemble" but it is not clear how a model like ACM-GPP-ET can solve this problem? Indeed, they are not based on the same parameters and set of equations so how to infer sensitivity of parameters from the complex model from sensitivity of parameters from intermediate complexity model? Likewise how calibration of intermediate complex model (especially in the case on the paper it is the opposite).

Thank you for your comment. It is quite correct that we should be clear in how we can propagate information using ACM-GPP-ET to other models. To inform better inform on this we have added a paragraph to the discussion section.

P32 L23-32: "The combination of ACM-GPP-ET, coupled to DALEC, and CARDAMOM provide multiple direct and indirect avenues for propagating information acquired using intermediate complexity models to complex state-of-the-art TEMs. ACM-GPP-ET and SPA directly share 5 parameters calibrated in this study (Table 2) plus a further 9 biophysical traits which where not calibrated in this study (Table 3). Moreover, all of the parameters calibrated in this study (Table 2) can be indirectly related to those used in SPA (and many other TEMs) e.g., NUE which is closely related to Vcmax, Jmax and foliar nitrogen, but also radiation absorption/reflectance as a function of LAI. Similarly, when ACM-GPP-ET is combined with DALEC and used within the CARDAMOM framework analyses such as those carried out by Bloom et al. (2016) (as is intended) retrieving information on carbon stocks, carbon allocation and residence times results in retrieval of ecologically relevant traits. These traits can be directly related to parameters found in most state-of-the-art TEMs equipped with a C-cycle. Such information should at a minimum provide information on spatial variation expected, and in the optimum situation inform on the exact magnitude of those parameters."

Likewise authors compare results for ACM-GPP-ET to different learning machine algorithms showing similar skill. But then what is the added values of such model compare to machine learning approaches? For me machine learning algorithms, are very well suited to interpolate informations but are difficult to use outside of their domain of training set.

Thank you for your comment. It is important to clarify the advantages of our approach over that of machine learning options. The addition of the new paragraph 4 in the introduction helps to highlight the advantages of a process-oriented approach which would not be possible via a machine learning approach. As you suggest, machine learning approaches are less able to extrapolate, i.e. make prediction outside of their calibration bounds which is essential in climate change related research. To this end Section 6.6 in the discussion has been expanded to include additional objectives of decadal predictions.

See paragraph 4 from introduction above.

P32 L22-32: "ACM-GPP-ET is well suited for ensemble modelling schemes due to its faster run-time, as shown in the MH-MCMC calibration process used here with SPA outputs used as training data. The parameter posteriors generated here (Figure A1) provide a starting point for full carbon cycle and water cycle analyses across regional to global domains. For instance, Bloom et al. (2016) have shown how an IC GPP model, ACM-GPP (Williams et al., 1997), combined with a carbon cycling model (DALEC; Williams et al., 2005), can be linked into a model-data fusion framework, CARDAMOM. CARDAMOM can, when combined with DALEC, retrieve probabilistic estimates of carbon stocks, fluxes and model parameters (including key unknowns such as photosynthate allocation to plant tissues and their residence times).

CARDAMOM can produce outputs across a domain at the resolution of input forcing (climate data, burned area) and observational constraints (satellite time series of LAI, biomass maps, soil C maps). The advantage of CARDAMOM is that it generates likelihoods for model initial conditions and parameter values that are consistent with climate forcing and domain observations from e.g. satellites, and their estimated errors. Currently CARDAMOM infers water limitations to C cycling through satellite observations of greenness alone. Because there is no coupling to a local water model, CARDAMOM cannot use modelled information on water balance, or independent observations such as surface soil moisture (e.g., Chen et al., 2018). Through using ACM-GPP-ET in CARDAMOM, it will be possible to assimilate new observational data related to water fluxes and state variables."

On the opposite model, based on process can be extrapolated. Then such model could be used for past or future climate conditions for instance. But ACM-GPP-ET is not a full vegetation model. In particular, it depend of LAI that should be prescribed and not calculated. So simulation are limited to conditions where LAI observations are available. So it would be interesting to know if the final objective is to include such kind of model in a TEM to be able to simulate the complete carbon cycle.

In summary the description of the model and its validation is correct. The model give very satisfying result. But what is really missing in the paper is what are the objective and perspective for such a model.

We are sorry for any confusion, we have not been clear as to the background of our modelling frameworks. We also were not clear about the intended coupling of ACM-GPP-ET to the DALEC C-cycle model, and calibration within the CARDAMOM model-data fusion framework. See various responses above for details.

**For more specific points:**

- One argument for the intermediate complexity model is the difficulty to have sub-daily climate forcing. I am not very convinced since, first of all most of global products are available at 3 to 6 hourly time steps. Then, even for data only available at daily time- steps, most of the TEM use weather generators to simulate a pseudo diurnal cycle. So this is not really a limit. An interesting question could be to know if intermediate complexity model directly based on daily data are performing better with daily data that more complex models coupled with a weather generator?

We have added a reference which specifically deals with the impact of downscaling meteorology on simulation errors (P2L32-P3L4). With regard to the availability of 3-6 hour information, we argue that there remains advantages to simulation at daily time step as this reduces to computational load substantially when model is used in analyses requiring large ensembles. Please see responses to reviewer 1 for details.

The paper partly reply to this point in the comparison between SPA and ACM-GPP-ET on fluxnet site where ACM-GPP-ET seems to perform a little better that SPA. However the difference is relatively small.

We thank the reviewer for this comment. We do not interpret the results as to mean that ACM-GPP-ET is actually better than SPA, particularly as the differences are small. Rather that errors in the input drivers and the observations underlie these results. See responses to reviewer 1 for the detailed response and modifications to the manuscript.

This first point brings me to a second one about FLUXNET. As ACM-GPP-ET is calibrated on SPA, it will obviously tend to have the same behaviour and then the same discrepancies when compare to FLUXNET (which is visible on Figure 7). So as suggested in the further opportunities it would be interesting to make a calibration of ACM-GPP-ET on observed GPP and ET from fluxnet to see how the calibration differ and how model improve compared to fluxnet.

The discussion text has been expanded to address this possibility in Section 6.6 "There are an array of next steps to undertake for further development both as a stand alone tool and as part of a coupled modelling framework along with DALEC and CARDAMOM. The ACM-GPP-ET parameters estimated here against SPA can be calibrated individually at FLUXNET2015 site (were sufficient biophysical information are available) to determine critical parameter variability to explain observed differences in fluxes. Driven with remotely sensed LAI ACM-GPP-ET could make global estimates of GPP, ET and WUE for direct comparison with outputs from FLUXCOM, GLEAM and CMIP5 model ensembles. As part of the CARDAMOM framework a site specific FLUXNET2015 analysis allows us to assess our ability to retrieve information on the whole carbon cycle across ecological and climate gradients within a data-rich environment, including key unknowns such as rooting depths which play a critical role in ecosystem resilience to drought. Such analyses provide the supporting frameworks needed to conduct global scale re-analyses and potentially near-term (next 12 months) and intermediate term (next 10 years) predictions with fully resolved uncertainties due to the propagation of ensembles."

- I am a little surprised by the calibration protocol making a simulation from 2001 to 2012 with rapid increase of CO2? Since I guess there is some prognostic variables it means that there is correlations between the successive years of simulation and then the rapid (and unrealistic) change in CO2 could lead to artefact in the simulation. So to explore the range of CO2 why not doing a series of simulations from 2001 to 2012 with different (but fixes) levels of CO2?

The calibration is done on individual days (i.e. the model is reset each day, including its prognostic variables) - the goal is to span a range of climate, atmospheric and ecological variability, and that explains the rising CO2 runs we used in SPA. We clarified the text as follows:

"Atmospheric CO2 concentration for each day was sampled from 300-450 ppm; the exaggerated CO2 range is to ensure that influence of increasing CO2 concentrations is contained within the calibration dataset."

- The way LAI is used in the calibration is not clear. It is stated that it is retrieved from DALEC and few lines after it is stated that CARDAMOM assimilate MODIS LAI ? So which LAI is used everyday to force ACM-GPP-ET ? And is the LAI from DALEC is retrieved from a previous simulation or done with a simulation with the same Era-iterim forcing (and CO2 increase) ? this point must be clarified

We apologise for the confusion caused in the description of the calibration and validation. These sections have been revised to clarify the situation. However a brief description is given here. ACM-GPP-ET is calibrated using the CARDAMOM system based on GPP and ET estimates from SPA. Both SPA and ACM-GPP-ET were driven with LAI and root biomass information extracted from a different CARDAMOM analysis which calibrated the DALEC C-cycle model:

"We used LAI and fine roots datasets Bloom et al. (2016) derived from MODIS LAI products, remotely sensed above ground biomass and ecological process knowledge (for details see; Bloom & Williams 2015."

- in equation 12 use P to define day length where different Pn,Pd,Pi represent different GPP limited term is not very appropriate !

The manuscript has been modified accordingly. See response to reviewer 1 for detailed response

**Description and validation of an intermediate complexity model for ecosystem photosynthesis and evapo-transpiration: ACM-GPP-ETv1**

Thomas Luke Smallman1,2 and Mathew Williams1,2

1School of GeoSciences, University of Edinburgh, Edinburgh UK 2National Centre for Earth Observations, University of Edinburgh, UK **Correspondence:** T.L. Smallman, t.l.smallman@ed.ac.uk

**Abstract.** Photosynthesis (gross primary production, GPP) and evapo-transpiration (ET) are ecosystem processes with global significance for climate, the global carbon cycle, climate, hydrologyand hydrological cycles, and a range of ecosystem services. The mechanisms governing these processes are complex but well understood. There is strong coupling between these processes, mediated directly by stomatal conductance and indirectly by root zone soil moisture content and its accessibility. This coupling

- 5 must be effectively modelled for robust predictions of earth system responses to global change. Yet, it is highly demanding to model leaf and cellular processes, like stomatal conductance or electron transport, with responses times of minutes, over decadal and global domains. Computational demand means models resolving this level of complexity cannot be fullyeasily evaluated for their parameter sensitivity, nor calibrated using earth observation datainformation through data assimilation approaches requiring large ensembles. To resolve this problemovercome these challenges, here we describe a coupled photosynthesis
- 10 evapo-transpiration model of intermediate complexity. The model reduces computational load and parameter numbers by operating at canopy scale and daily time steps. But by including Through the inclusion of simplified representation of key process interactions it retains sensitivity to variation in climate, leaf traits, soil states and atmospheric CO2. The new model is calibrated to match the biophysical responses of a complex terrestrial ecosystem model (TEM) of GPP and ET through a Bayesian model-data fusion process framework. The calibrated ACM-GPP-ET generates unbiased estimates of TEM GPP and ET, and captures
- 15 80-95 % percent of the sensitivity of carbon and water fluxes by the complex TEM. The ACM-GPP-ET model operates  $\sim$ 2200 timesthree orders faster than the complex TEM. Independent evaluation of ACM-GPP-ET at FLUXNET sites, using a single global parameterisation, shows good agreement with typical R2  $\sim$ 0.60 for both GPP and ET. This intermediate complexity modelling approach allows full Monte Carlo based quantification of model parameter and structural uncertainties, global scale sensitivity analyses for these processes, and is fast enough for use within terrestrial ecosystem model-data fusion frameworks
- 20 requiring large ensembles.

Copyright statement. TEXT

**1 Introduction**

Ecosystem photosynthesis and evaporation are key ecosystem fluxes, and their strong coupling generates important feedbacks between plant carbon and water cycles (Tuzet et al., 2003; Bonan and Doney, 2018). Ecosystem photosynthesis, or gross primary productivity (GPP) is generally the sole input of organic carbon into terrestrial ecosystems, ultimately determining

- 5 potential carbon accumulation rates. Ecosystem evaporation, or evapo-transpiration (ET), is the combination of plant mediated transpiration, soil surface evaporation and subsequent evaporation of rainfall intercepted by plant canopies. The dominant abiotic factors governing the magnitude and variability of GPP are temperature, absorbed photosynthetically active radiation (PAR) and CO2 which are strongly impacted by available leaf area index (LAI). Access to CO2 is controlled via leaf stomata, which provide the primary coupling point between GPP and the water cycle. Stomatal opening results in water loss via tran-
- 10 spiration creating a dependency on available water in the soil, but also accessible soil moisture, which is controlled by root biomass and its distribution through the soil profile. Thus, the soil-root interface is a second coupling point between the plant carbon and water cycles (Beer et al., 2009; Bonan and Doney, 2018). State-of-the-art terrestrial ecosystem models (TEMs) provide a mechanistic / process-oriented representation of the coupling between plant carbon and water cycles (e.g. Krinner et al., 2005; Oleson et al., 2010; Smallman et al., 2013; Harper et al., 2016) at leaf or even sub-leaf scale, resolving radiative transfer, stom-
- 15 atal conductance and electron transport. TEMs represent state-of-the-art knowledge on how ecosystems function, and are used to provide meaningful predictions of the responses by and feedbacks from the terrestrial land surface in response to changes in the Earth system (Bonan and Doney, 2018). Mechanistic models linking leaf-level photosynthesis (e.g., Farquhar and von Caemmerer, 1982; Collatz et al., 1991) and transpiration (e.g., Monteith, 1965) through models of stomatal regulation (Medlyn et al., 2011; Williams et al., 1996; Bonan et al., 2014) are well established. Scaling from leaf to canopy scale has grown in-
- 20 creasingly complex as the role of non-linear within-canopy variation of both abiotic (e.g., light, temperature, momentum, CO2 and H2O) and biotic (i.e. plant traits) factors on plant carbon-water relations has improved (e.g., Wang and Leuning, 1998; Buckley et al., 2013; Sun et al., 2014; Way et al., 2015; Coble et al., 2016; Scartazza et al., 2016; Nolan et al., 2017; Bonan et al., 2018).

However, the increasing complexity of TEMs makes them computationally expensive, which has several drawbackspresents new chal-

- 25 lenges. The slow speed of TEMs hinders their Many of the most complex TEMs are too slow for use in model-data fusion analyses which are reliant on massive ensemble simulations (e.g., Ziehn et al., 2012; Smallman et al., 2017). While effective and more computationally efficient alternative model-data fusion approaches are available they often reply on model code modifications, such as the creation of the model adjoint in variational approaches (e.g., Kuppel et al., 2012; Raoult et al., 2016), or model emulation often resulting in larger uncertainties in their posterior analysis (e.g., Fer et al., 2018). Moreover The complexity
- 30 of typical TEMs generally prevents a robust quantification of their uncertainties; it is very challenging computationally to determine the sensitivities of TEM model outputs to parameter variation. This hinders interpretation of model-data mismatch. Finally, there are major challenges in procuring sub-daily meteorological observations needed to drivers needed for TEMs away from meteorological stations this is a particularly acute problem in tropical regions. Thus, TEMs are generally run using statistical down-scaled climate reanalysis data. Such data necessarily contains, which contain a degree of errors which when propagated The

uncertainty generated when these errors are propagated into TEM GPP and ET estimates is significant compared to other sources of comparable to IC model error associated with simulating daily fluxes directly (Williams et al., 1997, 2001a). Thus, IC mod- els have been shown to have similar errors to TEM models but at lower computational cost (i.e. 1 time versus 24 time steps). Thus, there is considerable value in having less complex, fast-running models that simulate GPP and ET. The challenge here

5 is to produce a model *both* sufficiently mechanistic to represent the coupling between plant carbon and water cycles linking to ecophysiological processes and observations of key global unknowns (e.g. rooting depth), but also computationally fast enough to be integrated into model-data fusion schemes and to allow a full exploration of parameter-related uncertainties.

Photosynthesis is often estimated using physiologically realistic light,  $CO_2$  and temperature response functions (e.g., Jones, 1992; Williams et al., 1997), generally using vapour pressure deficit (VPD) as a proxy of moisture stress (e.g., Wang et al., 2017). Evaporation is

- 10 frequently estimated using simplified versions of the Penman-Monteith model, typically modelling plant stomatal regulation as a function of environmental drivers (e.g., Priestley and Taylor, 1972; Fisher et al., 2008). The impact of moisture limitations on both GPP and ET is commonly achieved through the use of VPD as a proxy (e.g., Mu et al., 2011; Wang et al., 2017) or using a single soil layer "bucket" (e.g., Martens et al., 2017). These simple and highly computationally efficient models can be driven by or combined with satellite based remotely sensed information within data assimilation schemes to constrain carbon or water fluxes at global scales with

[revised manuscript text omitted]

$$\quad \mathbf{P}_{NT} = \mathbf{LAI} \cdot \mathbf{N}_{fol} \cdot \mathbf{NUE} \cdot T_{adj} \tag{1}$$

Where  $T_{adj}$  describes a skewed normal distribution (scaling 0-1) with an optimum ( $T_{opt}$ ), maximum temperature ( $T_{max}$ ) and kurtosis (*Kurt*).

$$\mathbf{T}_{adj} = exp\left(log\left(\frac{\mathbf{T}_{max} - \mathbf{T}_{air}}{\mathbf{T}_{max} - \mathbf{T}_{opt}}\right) \cdot Kurt \cdot (\mathbf{T}_{max} - \mathbf{T}_{opt})\right) \cdot exp(Kurt \cdot (\mathbf{T}_{max} - \mathbf{T}_{opt}))$$
(2)